# Similarity-Enhanced Homophily for Multi-View Heterophilous Graph Clustering

## Abstract

With the increasing prevalence of graph-structured data, multi-view graph clustering, which aims to partition samples (represented as nodes in a graph) into distinct groups by leveraging complementary information from multiple graph views, has gained increasing popularity in various applications. While existing methods often employ a unified message passing mechanism to enhance clustering performance, this approach is less effective in heterophilous scenarios, where nodes with dissimilar features are connected. Our experiments demonstrate this by showing the degraded clustering performance as the heterophilous ratio increases. To address this limitation, a natural method is to conduct specific graph filters for graphs with specific homophilous ratio. However, this is inappropriate for unsupervised tasks due to the unavailable labels and homophilous ratios. Alternatively, we start from an observation showing that the implicit homophilous information may exist in similarity matrices even when the graph is heterophilous. Based on this observation, we explore a strategy that does not require prior knowledge of the homophilous or heterophilous, proposing a novel data-centric unsupervised learning framework, namely SiMilarity-enhanced Homophily for Multi-view Heterophilous Graph Clustering (SMHGC). By analyzing the relationship between similarity and graph homophily, we propose to enhance the homophily by introducing three similarity terms, *i.e.*, neighbor pattern similarity, node feature similarity, and multi-view global similarity, in a label-free manner. Then, a consensus-based inter- and intra-view fusion paradigm is proposed to fuse the improved homophilous graph from different views and utilize them for clustering. The state-of-the-art experimental results on both multi-view heterophilous and homophilous datasets highlight the effectiveness of using similarity for unsupervised multi-view graph learning, even in heterophilous settings. Furthermore, the consistent performance across semi-synthetic datasets with varying levels of homophily serves as further evidence of SMHGC's resilience to heterophily.

## 1 Introduction

Graph neural networks (GNNs) (Scarselli et al., 2008; Kipf & Welling, 2016b; Velickovic et al., 2019; Kipf & Welling, 2016a; Li et al., 2023) that can handle graph data by considering both node features and neighbor relations show impressive performance in various domains, such as social networks (Wei et al., 2023; Zhang et al., 2022; Su et al., 2022), recommendation systems (Deng et al., 2022; Xia et al., 2022a; Zhang et al., 2023) and molecules (Yu & Gao, 2022; Sun et al., 2022b; Li et al., 2022a). However, labeling the growing explosion of graph-structured data is often intricate and expensive. Multi-view graph clustering (MVGC) has recently been explored to address this problem under unsupervised settings. In MVGC, the features of samples are naturally from different views, and the relationships among samples, similarly, can be constructed from different views. The complementary and consensus information among different views with auxiliary graphs are, then, utilized for clustering. For example, O2MAC (Fan et al., 2020) captures the shared feature representation by designing a One2Multi GNN-based graph autoencoder. MVGC (Xia et al., 2022b) explores the cluster structure by training a graph convolutional encoder to learn the self-expression coefficient matrix. MCGC (Pan & Kang, 2021) learns a consensus graph to exploit both attribute content and graph structure information simultaneously.

Despite much progress made in this area, these methods based on GNNs typically rely on an implicit homophily assumption, *i.e.*, connected nodes often belong to the same class, as pointed out by Zhu et al. (2020); Ma et al. (2022). With this assumption, the message passing mechanism in GNNs can effectively aggregate the node information from the same class while disregarding the scrambled information, enabling the model to obtain class-distinguishable embedding for downstream tasks as substantiated by extensive empirical evidence (Song et al., 2022; Sun et al., 2022a; Jin et al., 2023; Han et al., 2023; Yu et al., 2023). Unfortunately, the graphs collected in reality often fail to fully satisfy the homophily assumption, which significantly limits the applicability of GNN-based MVGC methods. In fact, a more common graph is moderately or even mildly homophilous rather than a fully homophilous graph due to the general presence of non-homophilous (heterophilous) information in graphs. When it comes to such heterophilous graphs, the message passing mechanism aggregates the node information from different classes, hindering access to class discriminative embeddings. The empirical results shown in Fig. 1(a) demonstrate the challenge of heterophilous in the context of unsupervised clustering task, *i.e.*, the steady degrades of clustering performance with the increase of heterophilous ratio. Unlike supervised tasks with ground truth labels where the homophilous ratio could be estimated and then improved by the observation of training data, it is more difficult to design frameworks for unsupervised clustering on homophilous ratio agnostic graphs. Therefore, one key point for MVGC is *how to learn class discriminative embeddings on unsupervised tasks with unknown homophilous ratio*, which termed as multi-view heterophilous graph clustering (MVHGC).

Recently, several works have tried to address the heterophilous issue for MVGC. These works, generally, focus on two aspects. One is to utilize the pseudo labels predicted by multiple views to gradually improve the aggregated graphs, for example, DuaLGR (Ling et al., 2023) proposes a dual pseudo label guided framework. Another type tries to adaptively extract useful information by utilizing elaborate graph filters. For example, MCGC (Pan & Kang, 2021) and AHGFC (Wen et al., 2024) propose hybrid graph filters for learning node embedding. However, the first type relies on the robustness of pseudo labels, the bad quality of pseudo labels may lead to local optimums. The second type is limited due to the difficulty in determining which graph filter should be used for which specific graph, because of the unaccessible homophilous ratio.

Unlike these model-centric methods, we address this issue from a data-centric view, which could skip the limitations of previous works that rely on pseudo labels or GNN models (graph filters). Our observations on the benchmark data demonstrate the homophilous ratio of a graph could be potentially improved. As shown in Fig. 1(b), the homophilous ratio could essentially be improved by the proposed neighbor pattern similarity and feature similarity. This suggests that homophilous information can exist even in heterophilous graphs, a factor that has been overlooked by previous work.

Based on the observation, we propose our solution from a significantly effective yet generally underestimated perspective - *Similarity*. To avoid the limitations of GNNs on heterophilous graphs for clustering tasks, as described in Fig. 1 and Section 2.2, we propose three similarities, *i.e.*, neighbor pattern similarity, node feature similarity, and multi-view global similarity, to effectively improve the homophily of graph. Building on this concept, we propose a SiMilarity-enhanced homophily Multi-view Heterophilous Graph Clustering framework (SMHGC). In this framework, a robust similarity-enhanced homophilous graph can be obtained and iteratively updated with the optimization of the global similarity as well as the backpropagation of multi-view information, ultimately enhancing the subsequent message passing process and clustering results.

The experiments show that the proposed SMHGC achieves state-of-the-art (SOTA) results on both widely used homophilous multi-view graph datasets and heterophilous graph datasets. Moreover, when considering six semi-synthetic multi-view heteraphilous graph datasets with varying heterophilous ratios, SMHGC performs perfectly without any decrease in clustering performance even when the heterophilous ratio increases on semi-synthetic MVGC datasets. This contrasts sharply with previous studies, which showed a significant decline in clustering results. Especially on the semi-synthetic graph with heterophilous ratios greater than 70%, SMHGC significantly improves the normalized mutual information by over 30% compared to previous SOTAs. In addition, our ablation study further demonstrates the effectiveness of the proposed components.

The contributions of this paper can be summarized as follows:

- We propose three different similarities (neighbor pattern similarity, node feature similarity, and multi-view global similarity) to extract and fuse the homophilous information. The relationship

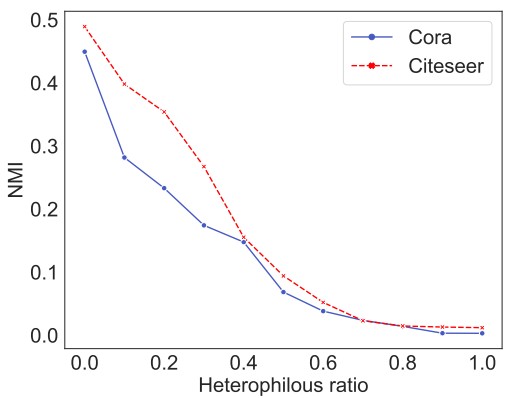 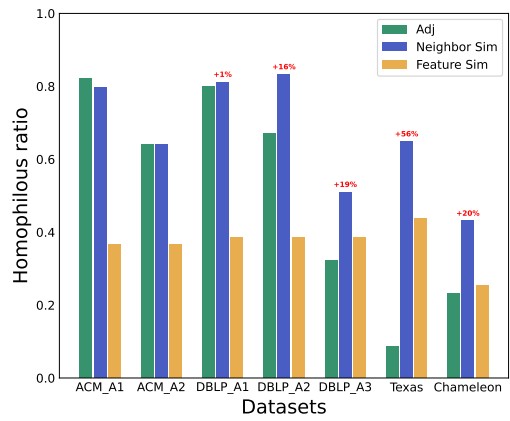

(a) On basic message passing mechanism, normalized mutual information (NMI) is heavily impacted by the heterophilous ratio. Graphs with varying heterophily ratios were generated by adjusting edge connections in a ACM base graph.

(b) Homophilous ratio on homophilous graphs (ACM and DBLP) and heterophilous graphs (Texas and Chameleon). Red is the surpassed value to green.

Figure 1: **(a) Observation 1**: Clustering performance decrease with the increase of heteropihlous ratio. **(b) Observation 2**: On heterophilous graph (Texas and Chameleon), homophilous ratio of neighbor pattern similarity (Definition 4) and feature similarity could be better than the original adjacent. Each entry in the dataset corresponds to a view in Table 4. For example, *ACM_A1* corresponds to $\mathcal{G}^1$ in Table 4. NMI here is a metric used to assess the similarity between the true class labels and the predicted labels, where higher values indicate better clustering performance. Homophilous Ratio (named HR), refers to the proportion of edges within the same class compared to the total number of edges in the graph, which is used to measure the homophily of the graph.

between homophily and our proposed similarity terms is analyzed, and we empirically demonstrate that the similarity can efficiently extract homophilous information in a label-free manner.

- Distinguishing from existing model-centric approaches, we propose a novel framework called SMHGC that processes heterophilous graphs before obtaining node embeddings. SMHGC can effectively utilize and optimize the homophilous information extracted by the proposed three similarity terms.

- Extensive experiments on both homophilous and heterophilous datasets demonstrate the superior performance of SMHGC. Moreover, we experimentally validate the feasibility and effectiveness of improved unsupervised learning performance by using similarity to extract homophily.

## 2 Preliminaries

### 2.1 Notations and Definitions

Let $\mathcal{G} = (\mathcal{V}, \mathcal{E})$ be an undirected graph, where $\mathcal{V} = \{x_i\}_{i=0}^N$ is the node set with node numbers $N = |\mathcal{V}|$, and $\mathcal{E} = \{e_{i,j} | 0 \le i, j < N\}$ is the edge set with self-loop. $\mathbf{X} \in \mathbb{R}^{N \times d}$ denotes the feature matrix of the nodes, where $x_i$ is the $d$-dimensional feature vector of node $i$. $\mathbf{A} \in \mathbb{R}^{N \times N}$ is the symmetric adjacency matrix of $\mathcal{G}$, where $a_{ij} = 1$ if there exists an edge between node $i$ and node $j$, otherwise $a_{ij} = 0$. Let diagonal matrix $\mathbf{D}$ represent the degree matrix of $\mathbf{A}$, *i.e.*, $\mathbf{D}_{ii} = \sum_j \mathbf{A}_{ij}$. Here we consider the normalized adjacency matrix $\mathbf{A}$ to be normailzed as $\tilde{\mathbf{A}} = \mathbf{D}^{-1}\mathbf{A}$. In the setting of MVGC, nodes and the relations among them can be seen from multiple views. Specifically, given $V$ different views with graphs $\{\mathcal{G}^v\}_{v=1}^V$ and features $\{\mathbf{X}^v\}_{v=1}^V$, which are assumed to be provided as input, the goal of MVGC is to partition the nodes into different classes. We formally define this task in Problem 1.

**Homophily and heterophily.** Different from homogeneity/ heterogeneity which describes the types of nodes and edges in a graph, homophily/ heterophily is a concept that describes the nature of edges in a graph (McPherson et al., 2001; Zheng et al., 2022). To make the distinction easier, we formally define them again as follows.

**Definition 1** (Homophily and heterophily (Zhu et al., 2020; Ma et al., 2022) ). *Let* $\mathbf{Y}$ *be the set of node labels, where* $y_i$ *denotes the label of node* $i$. *Let* $e_{i,j}$ *be an arbitrary edge connecting node* $i$ *and node* $j$ *in* $\mathcal{V}$. *Homophily describes* $y_i = y_j$ *for the edge* $e_{i,j}$, *and, conversely, it is termed heterophily.*

For a graph, it consists of only homophilous and non-homophilous edges (heterophilous edges). Based on this, we can define the homophilous information in a graph with the help of generalized edge as follows:

**Definition 2** (Generalized edge). *The generalized edge set* $\widetilde{\mathbf{E}}$ *is defined by transforming the inputs* $\mathbf{X}$ *and* $\mathbf{A}$ *through any transformation operation* $Tf(\cdot)$ *(e.g., initial inputs or multi-order aggregation):* $\widetilde{\mathbf{E}} = Tf(\mathbf{X}, \mathbf{A}) : \mathbb{R}^{N \times d} \times \mathbb{R}^{N \times N} \to \mathbb{R}^{N \times N}$. *Each element* $\widetilde{e}_{i,j}$ *in* $\widetilde{\mathbf{E}}$ *is a generalized edge that connects nodes* $i$ *and* $j$.

**Definition 3** (Homophilous information). *For an arbitrary generalized edge* $\widetilde{e}_{i,j} \in \widetilde{\mathbf{E}}$, *it exhibits homophily if and only if* $y_i = y_j$, *i.e., nodes* $i$ *and* $j$ *connected by it belong to the same cluster. The homophilous information is defined as the set of homophilous generalized edges.*

In recent works (Zhu et al., 2020; Lim et al., 2021), the ratio of homophilous edges, named homophily ratio, is used to measure the homophily of a given graph. The graph becomes strongly homophilous when this ratio closes to 1, and conversely, the graph is with strong heterophily (*i.e.*, weak homophily) when the ratio closes to 0. However, it is notable that this metric can only be used for evaluation but cannot contribute to the learning of models in unsupervised scenarios since this process relies on labels that are agnostic.

With the aforementioned notations, the problem of MVHGC can be formalized as follows.

**Problem 1** (Multi-view heterophilous graph clustering).
**Input:** *Graphs* $\{\mathcal{G}^v\}_{v=1}^V$ *encompassing both homophilous and heterophilous information and corresponding node feature matrices* $\{\mathbf{X}^v\}_{v=1}^V$ *from* $V$ *views.*
**output:** *The clustered node labels for the total* $N$ *nodes.*

## 2.2 Limitation of GNNs for Clustering on Heterophilous Graph

Generally, contemporary GNNs rely on a message passing mechanism in which each node updates itself by aggregating the embedding of its neighboring nodes and combining it with its own embedding (Xu et al., 2019). Specifically, the embedding update process of node $i$ at the $l$-th GNN layer can be expressed as:

$$m_i^l = \text{AGGREGATE}^l(\{h_u^{l-1}|u \in \mathcal{N}(i)\}); h_i^l = \text{UPDATE}^l(h_i^{l-1}, m_i^l), \tag{1}$$

where $h_i^l$ denotes the embedding of node $i$ at the $l$-th layer, $\mathcal{N}(i)$ represents the neighborhood of node $i$ and $m_i$ represents the messages aggregated from the neighborhood of node $i$. AGGREGATE($\cdot$) and UPDATE($\cdot, \cdot$) are defined as aggregation and update operations in the forward process. From Eq. (1), it can be seen that messages from neighborhood aggregation play a crucial role in the update of node $i$'s embedding. In homophilous neighborhoods, messages are clear and come from pure neighbors of the same class. While in heterophilous neighborhoods, mixed-class information weakens embedding distinguishability. We claim that this distinguishability also exists in unsupervised clustering task.

**Observation from basic message passing on synthetic datasets.** Fig. 1(a) demonstrates this by performing a basic two-order nonparametric message passing on two single-view datasets (Cora and Citeseer) respectively, showing that performance is greatly affected by heterophilous information. Therefore, directly aggregating messages from neighbors through a heterophilous graph may not be an effective choice.

**Observation from complex GNN on real-world datasets.** Furthermore, we explore the limitation illustrated by conducting a more complex GNN-based clustering baseline GAE (Kipf & Welling, 2016a) on real-world graphs from DBLP and ACM with different heterophilous ratios. From Table 2.2, we can conclude

Table 1: Clustering results of GAE on three views of DBLP and two views of ACM. The clustering results generally decrease with the increase of heterophilous ratio. The values in brackets mark the heterophilous ratio of the views in the dataset.

| Dataset | DBLP | | | ACM | |
|---|---|---|---|---|---|
| View | View 1 (0.20) | View 2 (0.33) | View 3 (0.68) | View 1 (0.18) | View 2 (0.36) |
| ACC% | 66.06 | 76.53 | 41.36 | 50.28 | 67.83 |
| NMI% | 41.23 | 63.43 | 10.65 | 26.53 | 45.06 |
| ARI% | 37.16 | 59.80 | 8.41 | 13.51 | 40.25 |
| F1% | 55.85 | 64.53 | 30.80 | 52.46 | 68.66 |

similar observations as Fig 1(a), *i.e.*, GAE generally performs worse on graphs with higher heterophilous ratio. It is worth noting that View 2 on DBLP slightly outperforms View 1, even though it has a higher heterophilous ratio. This suggests that View 2 has the potential to learn more effective partitions by considering factors beyond just the homophily present in the original graph, *e.g.,*, the similarity terms proposed in this study.

## 3 Proposed Framework

### 3.1 Overview of SMHGC

In this section, we present the SMHGC framework, focusing on three key similarities. Fig. 2 illustrates an overview of SMHGC. $V$ feature matrices ($\{\mathbf{X}^v\}_{v=1}^V$) and corresponding $V$ graphs ($\{\mathbf{A}^v\}_{v=1}^V$) are input. Then, in the homophilous information extraction module, the feature matrices and adjacent matrices are transformed into two similarity subspaces $\mathbf{Z}_a$ and $\mathbf{Z}_x$ respectively to construct two similarities, *i.e.*, neighbor pattern similarity ($\mathbf{A}_a$) and node feature similarity ($\mathbf{A}_x$) (Section 3.2). After that, in the intra-view fusion module, the global similarity matrix $\overline{\mathbf{H}}\overline{\mathbf{H}}^{\mathrm{T}}$ which implies multi-view consistency similarity information is introduced to measure and weight the aforementioned two similarities, so that the graph $\mathbf{S}^v$ obtained from the three similarities is able to absorb not only the homophilous information from neighbor pattern and node feature but also the multi-view consistency similarity information from different views. This extracted homophilous structure is then aggregated with node features to get intra-view embeddings (Section 3.3). Finally, multiple view-specific embeddings are fused under the guidance of consensus information to produce a better global embedding, $\overline{\mathbf{H}}$. Iteratively, the updated $\overline{\mathbf{H}}$ further aids in the generation and optimization of view-specific similarity graph $\mathbf{S}$ (Section 3.4). Finally, the optimized global embedding is fed into traditional clustering methods (*e.g.*, $K$-means) to obtain clustering results.

Overall, SMHGC utilizes similarity to extract and enhance the homophilous information that is implied in features and heterophilous graphs, and then they are weighted and fused under the guidance of the multi-view global similarity. Therefore, a similarity graph $\mathbf{S}^v$ with comprehensive homophilous information can be obtained, benefiting the following message passing process as demonstrated by the practical experiments in Fig. 1(a), *i.e.*, more homophilous information results in better clustering results after message passing.

### 3.2 Couple Similarity Enhanced Homophily

**Neighbor pattern similarity for homophilous information extraction.** For a graph, it consists of only homophilous and heterophilous information. Instead of directly using this graph to aggregate neighbor messages, our motivation is to extract the implied homophilous information from this graph, so the negative effect of heterophilous information can be effectively avoided. A natural question is *what is the homophilous information in a heterophilous graph?*

To answer the question, we investigate various prior studies. For example, He et al. (2023) demonstrated that neighbor patterns are the key factor for GNNs to improve performance. Moreover, some research suggests that 'good heterophily', where the nodes with the same label sharing similar neighbor patterns, can be exploited to achieve good performance (Song et al., 2023; Ma et al., 2022). These research provides us a

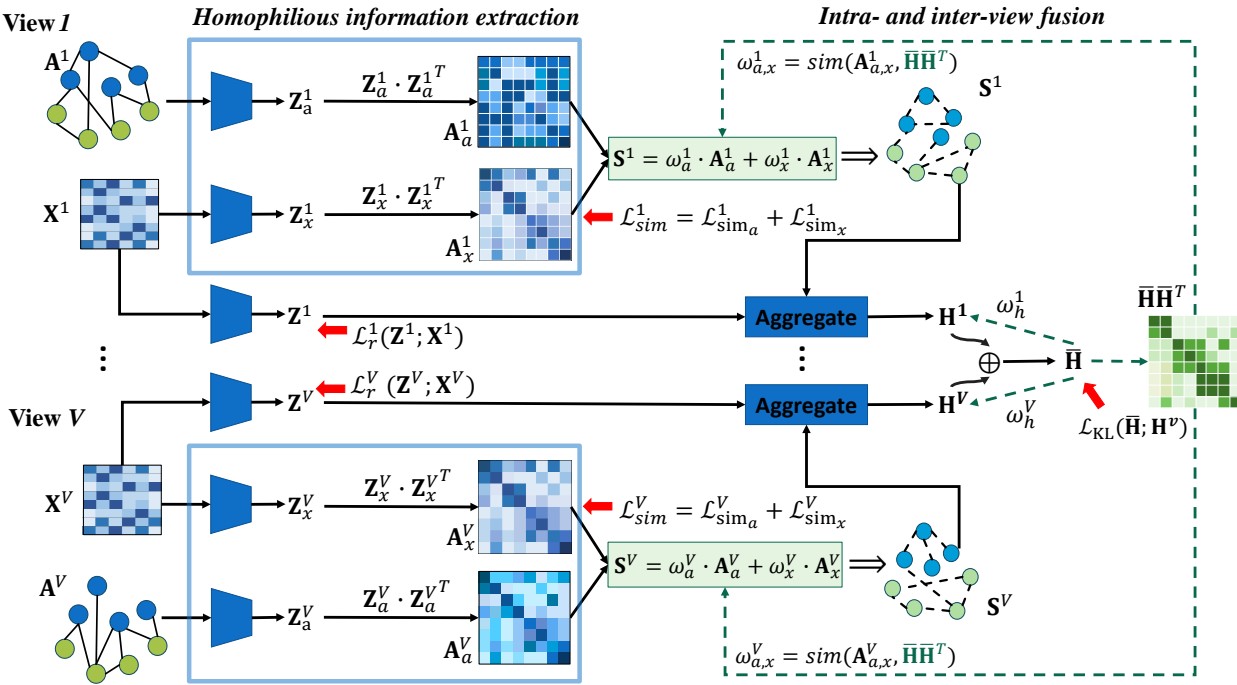

Figure 2: The proposed framework of SMHGC. It takes $V$ feature matrices ($\{\mathbf{X}^v\}_{v=1}^V$) and corresponding $V$ graphs ($\{\mathbf{A}^v\}_{v=1}^V$) as inputs. Then, homophilous information is extracted and optimized under the regularization of similarity terms ($\{\mathcal{L}_{sim}^v\}_{v=1}^V$). The homophilous graphs ($\{\mathbf{S}^v\}_{v=1}^V$) are then generated by infusing the homophilous information from both neighbor similarity matrix ($\mathbf{A}_a^v$) and feature similarity matrix ($\mathbf{A}_x^v$). Subsequently, the intra- and inter-view fusion module aggregates and fuses feature and homophilous information together to finally output a comprehensive embedding $\overline{\mathbf{H}}$.

support and possible solution to find homophilous information implied in heterophilous graphs. Following these previous works (Ma et al., 2022), we emphasize their observation in the following Proposition 1 :

**Proposition 1** (Good heterophily (Ma et al., 2022))**.** *In heterophilous graphs, if the neighborhood distribution of nodes with the same label is (approximately) sampled from a similar distribution and different labels have distinguishable distributions, then this heterophilous graph indicates good heteropihly.*

Regrettably, this notion of 'good heterophily', as inferred from the label information, is unsuitable for unsupervised MVHGC task. To generalize this proposition to unsupervised setting, we propose the hypothesis:

**Hypothesis 1.** *In node clustering tasks, the higher the similarity between two nodes in an ideal subspace, the greater the likelihood that they belong to the same cluster.*

The detailed discussion about Hypothesis 1 can be found in Appendix D.1 . With Hypothesis 1, the labels mentioned in Proposition 1 can be generalized to similarity, which implies that the nodes with similar neighbor patterns in the ideal subspace are homophilous nodes, so that to be used in the clustering task. More importantly, the target of learning homophilous information from a good heterophilous graph can be naturally changed to the target of learning a better projection that can project the input nodes to the ideal subspace which implies similarity information. Specifically, the neighborhood distribution of a node mentioned in Proposition 1 refers to a row in the adjacent matrix $\mathbf{A}$, and the cosine similarity of neighborhood distribution is naturally formulated as $\mathbf{A}\mathbf{A}^{\mathrm{T}}$ which we define as neighbor pattern similarity.

**Definition 4** (Neighbor pattern similarity)**.** *Given adjacency matrix $\mathbf{A}$, each row in $\mathbf{A}$ describes the neighbor pattern of the node. Therefore, we define $\mathbf{A}\mathbf{A}^{\mathrm{T}}$ as neighbor pattern similarity.*

Therefore, it becomes achievable to learn the homophilous information implied in good heterophilous graphs by the regularization of neighbor pattern similarity.

For an intuitive understanding, from the neighborhood aggregation perspective, neighbor pattern similarity implies 2-hop structural information, which provides a possibility to receive more homophilous edges compared to the initial heterophilous adjacency matrix. The experimental results shown in Fig. 3 also demonstrate that the neighbor pattern similarity may capture homophilous information implied in heterophilous graph.

Following the analysis, for acquiring homophilous information, we employ deep encoders to initially conduct subspace learning for node neighbors. This process is then refined through multi-view consensus, facilitated by a regularization term in Eq. (5) that captures neighbor pattern similarity. Specifically, each row of the given graph $\mathbf{A}$ implies a neighbor pattern of a node, so $\mathbf{A}$ is fed into an encoder $f_a$ instantiated with a multi-layer perception (MLP) with learnable parameters $\theta_a$, and then outputs a low-dimensional representation:

$$\mathbf{Z}_a = f_a(\mathbf{A}; \theta_a), \tag{2}$$

where $\mathbf{Z}_a \in \mathbb{R}^{N \times d'}$ aims to represent the potential neighbor pattern in ideal subspace.

Subsequently, we propose the neighbor pattern similarity regularization term $\mathcal{L}_{sim_a}$ to encourage the encoder $f_a$ to focus on learning neighbor pattern similarity information of the input graph:

$$\mathcal{L}_{sim_a} = l_s(\mathbf{Z}_a \mathbf{Z}_a^{\mathrm{T}}, \mathbf{A}\mathbf{A}^{\mathrm{T}}), \tag{3}$$

where $l_s(\cdot, \cdot)$ is loss calculation function, instantiated here as the Mean Squared Error (MSE) loss. In this similarity loss, $\mathbf{A}\mathbf{A}^{\mathrm{T}}$ is the neighbor pattern similarity information of the input graph. Consequently, by utilizing this loss, we aim to guide the encoder $f_a$ to learn as much information as possible regarding neighbor pattern similarity among nodes, capturing the potential homophily within the given heterophilous graph.

**Comparison with the definition of random walk $\mathbf{A}^n$.** It is notable that the definition on neighbor pattern similarity shares similar formula with traditional random walk $\mathbf{A}^n$ that defines $n$-steps walks on the graph. However, our focus on $\mathbf{A}\mathbf{A}^{\mathrm{T}}$ serves a specific purpose and interpretations in the context of this research. The definition of neighbor pattern similarity aims to address the challenge of finding homophilous information in heterophilous graphs, particularly in the context of MVHGC, and the formula of neighbor pattern similarity $\mathbf{A}\mathbf{A}^{\mathrm{T}}$ is derived from the cosine similarity of neighborhood distribution between nodes. For example, the element $(\mathbf{A}\mathbf{A}^{\mathrm{T}})_{ij}$ represents the number of common neighbors between nodes $i$ and $j$. Larger value of $(\mathbf{A}\mathbf{A}^{\mathrm{T}})_{ij}$ means nodes $i$ and $j$ have more common neighbors, therefore share larger neighbor pattern similarity. This measure is particularly useful for identifying nodes that share similar local structures. Unlike $\mathbf{A}^n$, which captures global connectivity over longer paths, $\mathbf{A}\mathbf{A}^{\mathrm{T}}$ focuses on local neighborhood information. This makes it a more suitable measure for identifying homophilous nodes in heterophilous graphs.

**Node feature similarity for homophilous information extraction.** On the other hand, the node feature $\mathbf{X}$ describes the inherent properties of the nodes, and these properties can also reflect the homophilous information between the nodes. Therefore, in addition to relying on the neighbor relations within $\mathcal{G}$, more homophilous information can be explored through the similarity information within the node features. Similar to neighbor pattern similarity, an encoder $f_x(\mathbf{X}; \theta_x) = \mathbf{Z}_x$, where $\mathbf{Z}_x \in \mathbb{R}^{N \times d'}$, is trained using the node feature similarity regularization term $\mathcal{L}_{sim_x}$:

$$\mathcal{L}_{sim_x} = l_s(\mathbf{Z}_x \mathbf{Z}_x^{\mathrm{T}}, \mathbf{X}\mathbf{X}^{\mathrm{T}}). \tag{4}$$

Finally, the couple similarity-based regularization loss for extracting homophilous information can be expressed as:

$$\mathcal{L}_{sim} = \mathcal{L}_{sim_a} + \mathcal{L}_{sim_x} = l_s(\mathbf{Z}_a \mathbf{Z}_a^{\mathrm{T}}, \mathbf{A}\mathbf{A}^{\mathrm{T}}) + l_s(\mathbf{Z}_x \mathbf{Z}_x^{\mathrm{T}}, \mathbf{X}\mathbf{X}^{\mathrm{T}}). \tag{5}$$

Constructed upon the concepts of the two similarity terms and Hypothesis 1, the autoencoder design for projecting $\mathbf{A}$ to $\mathbf{Z_a}$ by $f_a(\cdot)$ and $\mathbf{X}$ to $\mathbf{Z_x}$ by $f_x(\cdot)$ has two main goals: (1) Multi-view collaborative optimization: the backpropagation of final multi-view clustering loss ($\mathcal{L}_{KL}$) introduces clustering information from

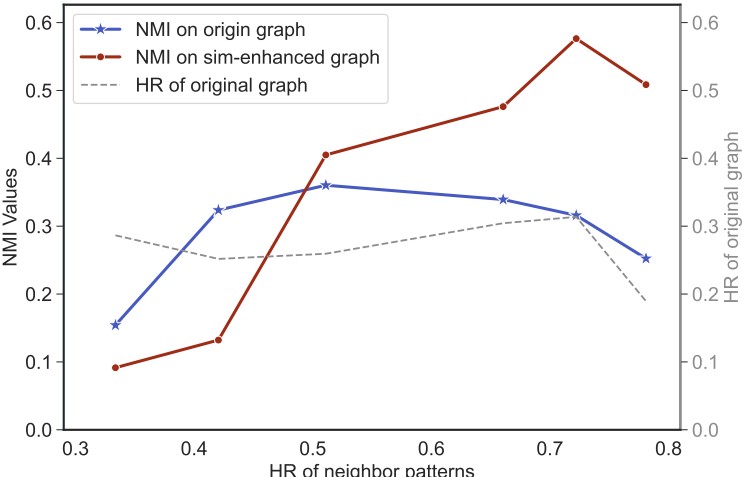

Figure 3: Left of Y-axis is the clustering performance (NMI). X-axis denotes synthesis datasets with increased 'good heterophily' constructed from ACM. To construct the synthesis dataset, the homophilous ratio of the original graph is kept low (as the gray dotted line shows), instead, we gradually increase 'good heterophily' (HR of neighbor patterns) information (more details can be found in Appendix E.3) . The original graph and sim-enhanced graph are fed into a parameter-free message passing layer to get aggregated node embedding, and $K$-means is conducted to obtain clusters evaluated by NMI.

other views (Zhu et al., 2024; Xu et al., 2021), achieving multi-view collaborative optimization for each single view's similarity matrices. (2) Learn a better subspace: as stated in Hypothesis 1, the similarity matrices are expected to be constructed in an ideal subspace. The projection of $\mathbf{Z}_x\mathbf{Z}_x^T$ and $\mathbf{Z}_a\mathbf{Z}_a^T$ can not only preserve most important similarity information (implemented by regularization term $\mathcal{L}_{sim}$) and feature information (by $\mathcal{L}_r$), but also obtain the guidance from other views (Peng et al., 2018; Baldi, 2012), resulting in the possibility of learning an ideal subspace.

**How the couple similarity enhances clustering performance.** The neighbor pattern similarity defined in Definition 4 enables the extraction of both homophilous information and 'good heterophily' from the input graph in a label-free manner. On the one hand, it involves a 2-hop aggregation of the adjacency matrix, allowing for the comprehensive mining of homophilous information within it. On the other hand, through similarity computation, similar neighboring patterns can be efficiently explored, facilitating the identification of 'good heterophily'. In addition, feature similarity extracts homophilous information from feature space.

In this subsection, we empirically explore how the neighbor pattern similarity and feature similarity enhance the clustering performance, therefore substantiating Hypothesis 1 in the meantime. Fig. 1(a) suggests that the message passing on heterophilous graph leads to poor clustering performance. Furthermore, as the blue line depicted in Fig. 3, the increase of homophilous information in a heterophilous graph still cannot increase the clustering performance via a message passing layer. This is mainly because of the unchanged HR of original graph, although the homophilous information is added by increasing the HR of neighbor patterns. In comparison, we construct an enhanced graph from couple similarity (sim-enhanced graph, denoted as $S$ below), which directly combines the information obtained from neighbor pattern similarity and feature similarity via weighted sum:

$$S = \omega_x \mathbf{X}\mathbf{X}^{\mathrm{T}} + \omega_a \mathbf{A}\mathbf{A}^{\mathrm{T}},$$

where $\mathbf{X}\mathbf{X}^{\mathrm{T}}$ and $\mathbf{A}\mathbf{A}^{\mathrm{T}}$ are the targets of our proposed regularization terms, and $\omega_x$ and $\omega_a$ are set as the homophilous ratio of feature similarity and neighbor pattern similarity respectively. The results in Fig. 3 are shown in red line: the increase of 'good heterophily' information leads to better clustering performance on sim-enhanced graph as inputs of a message passing layer.

This observation precisely explains why the proposed SMHGC maintains superior clustering performance shown in Fig. 4(a) and Fig. 4(b). The results presented in Fig. 3 suggest that *the couple similarity enhanced*

*graph could capture richer homophilous information from heterophilous graph, including 'good heterophily' (denoted as HR of neighbor patterns), compared to the initial input, leading to better clustering performance.*

### 3.3 Global Similarity Guided Intra-View Homophily Fusion and Aggregation

In unsupervised tasks, the inaccessibility of labels and homophily may lead to a lack of robustness when extracting homophilous information. Therefore, in this section, we address the challenge that *how to obtain more robust homophilous information*, and seek the guidance of consensus from different views.

**How to fuse node feature and neighbor pattern similarities.** In this section, we illustrate our intuition on how to fuse the two similarity terms by focusing on three questions. **Question 1**: Why is it required to compute the weights for two similarities? The node features and neighbor patterns in each view's graph $\mathcal{G}^v$ may imply different homophilous information, which contributes differently to the downstream task. To generate a robust homophilous graph, we design a global similarity guided intra-view homophily fusion strategy, aiming to assign weights to the extracted homophilous information based on their relevance to the final task and multi-view global similarity.

**Question 2**: Why is $\overline{\mathbf{H}}$ used to compute the weights of the two similarities? Without labels, it is difficult to directly determine the contribution of information to the final task. An alternative approach is inspired by clustering algorithms like $K$-means (MacQueen, 1967), where the distance between samples and their cluster centroids dictates their final assignments. In other words, the distance between samples plays a basis role for final assignments. Similarly, the fused embedding from all views (denoted as $\overline{\mathbf{H}}$), which is used for final assignments, captures the basis that is most aligned with the downstream tasks.

Therefore, it is natural to use the global similarity matrix obtained from $\overline{\mathbf{H}}$ to evaluate the homophilous information of node features and neighbor patterns.

**Question 3**: How is $\overline{\mathbf{H}}$ used to compute the weights of the two similarities? Specifically, let the homophilous information from node features and neighbor patterns be denoted as $\mathbf{A}_x = \mathbf{Z}_x \mathbf{Z}_x^{\mathrm{T}}$ and $\mathbf{A}_a = \mathbf{Z}_a \underline{\mathbf{Z}_a}^{\mathrm{T}}$, respectively. Their contribution to the task objective can be assessed by evaluating their similarity to $\overline{\mathbf{H}}$:

$$(\omega_x, \omega_a) = \text{norm}(sim(\mathbf{A}_x, \overline{\mathbf{H}}\overline{\mathbf{H}}^{\mathrm{T}}), sim(\mathbf{A}_a, \overline{\mathbf{H}}\overline{\mathbf{H}}^{\mathrm{T}})) \tag{6}$$

where $\text{norm}(x,y) := ((\frac{x}{\max(y,x)})^\rho, (\frac{y}{\max(y,x)})^\rho)$ denotes normalization operation, $sim(\cdot, \cdot)$ denotes the similarity calculation function, which is instantiated as cosine similarity in this work, and $\overline{\mathbf{H}}\overline{\mathbf{H}}^{\mathrm{T}}$ is the global similarity matrix. Based on this, $\mathbf{A}_x^v$ and $\mathbf{A}_a^v$ in $v$-th view can be properly fused to generate a homophilous graph as follows:

$$\mathbf{S}^v = \omega_x^v \mathbf{A}_x^v + \omega_a^v \mathbf{A}_a^v. \tag{7}$$

To accommodate the discrete nature of graph, we discretize the dense $\mathbf{S}^v$. Specifically, let $U_i^v$ be the set of the top $k$ largest elements in $\mathbf{S}_{i,:}^v$, where $k$ is a hyperparameter, thus $\mathbf{S}^v$ can be discretized as:

$$s_{ij}^v = \begin{cases} 1, & \text{if} \quad s_{ij}^v \in U_i^v, \\ 0, & \text{otherwise.} \end{cases} \tag{8}$$

**Aggregate node feature and extracted graph via GNN.** Compared to the neighbor relations in a graph composed of solely homophilous and heterophilous edges, the node features encompass a variety of information that can be leveraged for obtaining distinguishable embeddings. Therefore, the node features are required to be compressed in a latent space. In this work, autoencoders are employed to compress the inherent distinguishability of these node features. Let $f_{\phi^v}^v$ and $g_{\xi^v}^v$ be the encoder and decoder with parameters $\phi^v$ and $\xi^v$, respectively. Then, the node feature $\mathbf{X}^v$ in the $v$-th view can be reconstructed as $\hat{\mathbf{X}}^v = g_{\xi^v}^v(\mathbf{Z}_f^v) = g_{\xi^v}^v(f_{\phi^v}^v(\mathbf{X}^v))$, where $\mathbf{Z}_f^v$ is the latent representation. Based on this, the reconstruction loss function used to train the autoencoder is:

$$\mathcal{L}_r^v = l_r(\hat{\mathbf{X}}^v, \mathbf{X}^v) = l_r(g_{\xi^v}^v(f_{\phi^v}^v(\mathbf{X}^v)), \mathbf{X}^v), \tag{9}$$

where $l_r(\cdot, \cdot)$ is the loss function, which can be instantiated as a cross-entropy loss.

Given the construction of the graph $\mathbf{S}^v$ by associating homophilous information and inspired by Wu et al. (2019), we opt to eliminate redundant parameters in the graph convolution process. Instead, we directly convolve $\mathbf{S}^v$ with node feature $\mathbf{Z}_f^v$, which not only reduces model complexity but also enhances its generalization ability. Furthermore, we implement the aggregation in the form of residuals, preserving information for each order of operation. This simplified GNN is expressed as:

$$
\begin{aligned}
\mathbf{H}^v &= \mathbf{h}^{v,0} + \mathbf{h}^{v,1} + \cdots + \mathbf{h}^{v,order}, \\
\mathbf{h}^{v,order} &= (\mathbf{S}^v)^{order}\mathbf{Z}_f^v,
\end{aligned}
\tag{10}
$$

where $\mathbf{H}^v$ is the embedding of the $v$-th view, $\mathbf{h}^{v,0} = \mathbf{Z}_f^v$, and $order$ is a hyperparameter that is designed to control the order of aggregation. Note that the conducted aggregation in Eq. (10) can be replaced by any complicated GNNs.

### 3.4 Global Consensus Based Inter-View Fusion

In multi-view tasks, each view can contribute different information to the downstream task to varying degrees. To fully exploit the complementarity of the individual views, it is natural to fuse the embeddings from all views. However, the quality of information may vary across views, necessitating the assignment of appropriate weights to each view during fusion. Given the absence of label information, an alternative approach is to use inter-view consensus for evaluation. Specifically, we utilize the fused embedding $\overline{\mathbf{H}}$ from the current iteration to evaluate and score the embedding $\mathbf{H}^v$ obtained from each view. Based on this, we determine the weight of each view for fusion. Ultimately, the weighted fused embedding is updated:

$$
\overline{\mathbf{H}} = \sum_{v=1}^{V} \omega_h^v \mathbf{H}^v, \quad \omega_h^v = \left( \frac{score^v}{\max\left(score^1, score^2, \cdots, score^V\right)} \right)^{\rho_h},
\tag{11}
$$

where $score^v$ can be obtained from the evaluation function, $score^v = metric(\mathbf{H}^v, \overline{\mathbf{H}})$ and $metric(\cdot, \cdot)$ is instantiated as cosine similarity function in this work. The hyperparameter $\rho_h$ is applied to adjust the smoothing or sharpening of the view weights. Notably, the update of $\overline{\mathbf{H}}$ will in turn aid the generation of view-specific similarity graph $\mathbf{S}^v$ as the iteration proceeds. With $\overline{\mathbf{H}}$, sample clustering algorithms, such as $K$-means, can be implemented to obtain the final assignment.

### 3.5 Objective Function

Overall, the objective function of SMHGC comprises three parts: similarity regularization loss term of $V$ views $\sum_V \mathcal{L}_{sim}^v$ (Eq. (5)), reconstruction loss of $V$ views $\sum_V \mathcal{L}_r^v$ (Eq. (9)), and Kullback-Leibler (KL) divergence loss $\mathcal{L}_{kl}$:

$$
\mathcal{L} = \gamma_{sim} \sum_V \mathcal{L}_{sim}^v + \gamma_r \sum_V \mathcal{L}_r^v + \mathcal{L}_{kl},
\tag{12}
$$

where $\gamma_{sim}$ and $\gamma_r$ are hyperparameters to balance each loss. The KL divergence loss ($\mathcal{L}_{kl}$) is a commonly used clustering loss applied to facilitate the model to obtain consensus embedding following the previous works (Xie et al., 2016a; Ren et al., 2024). Specifically, let $q_{ij}^v \in \mathbf{Q}^v$ be the soft cluster assignment that describes the possibility of node $i$ belonging to cluster centroid $j$ in $v$-th view and $\mathbf{P}$ be the sharpen target distribution, the loss can be expressed as (more details can be found in Appendix D.1 ):

$$
\mathcal{L}_{kl} = \mathrm{KL}(\overline{\mathbf{P}}\|\overline{\mathbf{Q}}) + \sum_{v=1}^{V} \mathrm{KL}(\overline{\mathbf{P}}\|\mathbf{Q}^v),
\tag{13}
$$

where $\overline{\mathbf{Q}}$ and $\overline{\mathbf{P}}$ represent the soft and target distribution of $\overline{\mathbf{H}}$, respectively. With Eq. (23), we expect the first term to enhance the discriminability of the final embedding $\overline{\mathbf{H}}$, followed by the second term to encourage the soft distribution of each view to fit the distribution of $\overline{\mathbf{H}}$, and thus to exploit the inter-view consistency. Finally, the random initialized parameters in SMHGC are end-to-end trained by the objective function.

**Decouple the strategies for handling Low homophilous neighbor pattern similarity in SMHGC.** We discuss why SMHGC still can work when low homophilous neighbor pattern similarity exists. Specifically, SMHGC incorporates several strategies: 1) SMHGC conducts **subspace learning** to integrate neighbor pattern similarity as a regularization term, optimizing the subspace through both this similarity and losses from other views, thus reducing dependency on high homophily. Additionally, 2) **multi-view collaborative training** ensures parameters across different views are optimized together, enhancing the robustness of the subspace learning against low homophily. Furthermore, 3) **global similarity guided intra-view fusion** leverages multi-view consensus to assess and improve the homophilous ratio of learned similarities, ensuring that even if individual views have low homophily, the overall system can still perform well. Moreover, 4) **global consensus-based inter-view fusion** can adjust the influence of poorly performing views in the final embedding, ensuring that these views do not negatively impact the overall clustering quality. Lastly, 5) **dynamically weigh the contributions of different views** based on their reliability is employed by SMHGC to further improve the robustness and accuracy of the clustering results.

## 4 EXPERIMENTS

### 4.1 Experimental Settings

**Datasets.** To evaluate the effect of SMHGC, we conduct extensive experiments on ten datasets, including two real-world homophilous graph datasets (ACM[1] and DBLP[2]), two real-world heterophilous graph datasets (Texas[3] and Chameleon (Rozemberczki et al., 2021)) and six semi-synthetic graph datasets generated from ACM (Ling et al., 2023). More details about the datasets and implementation details can be found in the Appendix. The implementation of SMHGC will be published.

**Comparison methods.** The following baselines are considered: VGAE (Kipf & Welling, 2016a) is a single-view method. O2MAC (Fan et al., 2020), MvAGC (Lin & Kang, 2021), MCGC (Pan & Kang, 2021), MVGC (Xia et al., 2022b), DuaLGR (Ling et al., 2023) and BTGF (Qian et al., 2024) are six deep MVGC methods. The results from VGAE and O2MAC on ACM and DBLP are obtained from the best in the literature, the others are conducted five times and the average results with standard deviations are reported in Table 2. For MVGC (Xia et al., 2022b), we use the original version without feature augmentation technique for fair comparison.

**Evaluation metrics.** In order to evaluate the final clustering performance, this study adopts four commonly used metrics, including normalized mutual information (NMI), adjusted rand index (ARI), accuracy (ACC) and F1-score (F1), following previous works (Pan & Kang, 2021; Xia et al., 2022b; Chen et al., 2022).

### 4.2 Overall Performance

Table 2 presents the clustering performance of all compared methods on four real-world graph datasets. Notably, SMHGC demonstrates competitive ability with the baselines on homophilous datasets, including ACM and DBLP. Specifically, our method outperforms the best baseline DuaLGR on ACM, with NMI, ARI, ACC, and F1 improving by 12.6%, 6.7%, 2.0%, and 2.0%, respectively. On two heterophilous datasets, *i.e.*, Texas and Chameleon, our method exhibits excellent results compared to other traditional methods that rely on the homophily assumption. For instance, considering Texas, whose homophilous ratio *hr* is only 0.09, the best result from VAGE achieves only 55.3% accuracy, while SMHGC achieves significantly higher accuracy, reaching up to 71.3%. This underscores the outstanding performance of SMHGC on heterophilous graphs, leveraging similarity. It also highlights the limitations of existing traditional methods when confronted with heterophilous graphs. Additionally, Fig. 4(a) and Fig. 4(b) illustrate the partial clustering performance of SMHGC and other baselines on the six semi-synthetic ACM datasets with varying heterophilous ratios ranging from 0.5 to 1.0. The performance of these baselines deteriorates as the heterophilous ratio increases, indicating a decline in performance with the increase in heterophilous information. Notablly, the clustering

---

[1]https://dl.acm.org/

[2]https://dblp.uni-trier.de/

[3]http://www.cs.cmu.edu/afs/cs.cmu.edu/project/theo-11/www/wwkb

Table 2: The clustering results of SMHGC on two homophilous graph datasets and two heterophilous graph datasets. The best results are highlighted in bold, and second best are underlined.

| Methods / Datasets | ACM (*hr* 0.82 & 0.64) | | | | DBLP (*hr* 0.80 & 0.67 & 0.32) | | | |
|---|---|---|---|---|---|---|---|---|
| | NMI% | ARI% | ACC% | F1% | NMI% | ARI% | ACC% | F1% |
| VGAE (2016a) | 49.1 | 54.4 | 82.2 | 82.3 | 69.3 | 74.1 | 88.6 | 87.4 |
| O2MAC (2020) | 69.2 | 73.9 | 90.4 | 90.5 | 72.9 | 77.8 | 90.7 | 90.1 |
| MvAGC (2021) | 57.8 | 60.1 | 84.4 | 84.6 | 67.9 | 74.6 | 89.1 | 89.1 |
| MCGC (2021) | 70.9 | 76.6 | 91.5 | 91.6 | 72.2 | 77.5 | 90.4 | 89.8 |
| MVGC (2022b) | 62.4 | 61.4 | 83.1 | 82.2 | 47.3 | 45.4 | 71.8 | 69.4 |
| R$^2$FGC (2023) | 72.4 | 78.7 | 92.4 | 92.5 | 50.8 | 56.3 | 81.0 | 80.5 |
| MFCGC (2024) | 77.0 | 81.8 | 93.6 | 93.1 | **79.0** | **84.3** | **93.5** | **93.1** |
| DIAGC (2024) | 71.6 | 77.0 | 91.8 | 91.8 | 78.3 | 83.7 | 93.3 | 92.8 |
| DuaLGR (2023) | 73.1 ± 1.1 | 78.9 ± 0.7 | 92.5 ± 0.2 | 92.5 ± 0.2 | 75.4 ± 0.0 | 81.6 ± 0.0 | 92.3 ± 0.0 | 91.8 ± 0.0 |
| BTGF (2024) | 68.7 ± 0.0 | 73.6 ± 0.0 | 90.4 ± 0.0 | 90.5 ± 0.0 | 45.8 ± 0.0 | 49.0 ± 0.0 | 74.6 ± 0.0 | 73.6 ± 0.0 |
| SMHGC (ours) | **81.1 ± 4.1** | **83.2 ± 5.2** | **93.9 ± 2.0** | **93.9 ± 2.0** | 76.2 ± 0.8 | 81.9 ± 0.2 | 92.4 ± 0.2 | 91.8 ± 0.2 |

| Methods / Datasets | Texas (*hr* 0.09 & 0.09) | | | | Chameleon (*hr* 0.23 & 0.23) | | | |
|---|---|---|---|---|---|---|---|---|
| VGAE (2016a) | 12.7 ± 4.4 | 21.7 ± 8.4 | 55.3 ± 1.8 | 29.5 ± 3.1 | 15.1 ± 0.7 | 12.4 ± 0.6 | 35.4 ± 1.0 | 29.6 ± 1.7 |
| O2MAC (2020) | 8.7 ± 0.8 | 14.6 ± 1.8 | 46.7 ± 2.4 | 29.1 ± 2.4 | 12.3 ± 0.7 | 8.9 ± 1.2 | 33.5 ± 0.3 | 28.6 ± 0.2 |
| MvAGC (2021) | 5.4 ± 2.8 | 1.1 ± 4.1 | 54.3 ± 2.6 | 19.8 ± 5.1 | 10.8 ± 0.8 | 3.3 ± 1.7 | 29.2 ± 0.9 | 24.3 ± 0.5 |
| MCGC (2021) | 12.7 ± 2.9 | 12.9 ± 3.8 | 51.9 ± 0.9 | 32.5 ± 1.8 | 9.5 ± 1.3 | 5.9 ± 2.7 | 30.0 ± 2.0 | 19.1 ± 0.8 |
| MVGC (2022b) | 8.1 ± 3.3 | 7.8 ± 3.1 | 41.8 ± 2.6 | 28.4 ± 3.1 | 12.6 ± 0.3 | 5.1 ± 0.6 | 32.8 ± 0.4 | 26.9 ± 0.5 |
| DuaLGR (2023) | 32.6 ± 0.5 | 26.0 ± 0.6 | 54.3 ± 0.3 | 46.8 ± 0.3 | 19.5 ± 1.0 | 16.0 ± 0.6 | 41.1 ± 0.8 | 37.7 ± 1.5 |
| BTGF (2024) | 22.7 ± 0.0 | 20.5 ± 0.0 | 58.5 ± 0.0 | 35.1 ± 0.0 | 17.2 ± 0.0 | 11.5 ± 0.0 | 35.8 ± 0.0 | 30.7 ± 0.0 |
| AHGFC (2024) | 39.3 ± 5.6 | 45.3 ± 3.1 | 70.3 ± 1.0 | 45.7 ± 4.5 | **21.5 ± 0.4** | **16.2 ± 0.7** | **43.2 ± 0.3** | **41.6 ± 1.1** |
| SMHGC (ours) | **41.8 ± 1.1** | **46.9 ± 3.2** | **71.3 ± 0.8** | **49.8 ± 2.3** | 20.0 ± 1.3 | 15.1 ± 1.0 | 42.1 ± 0.8 | 41.3 ± 0.9 |

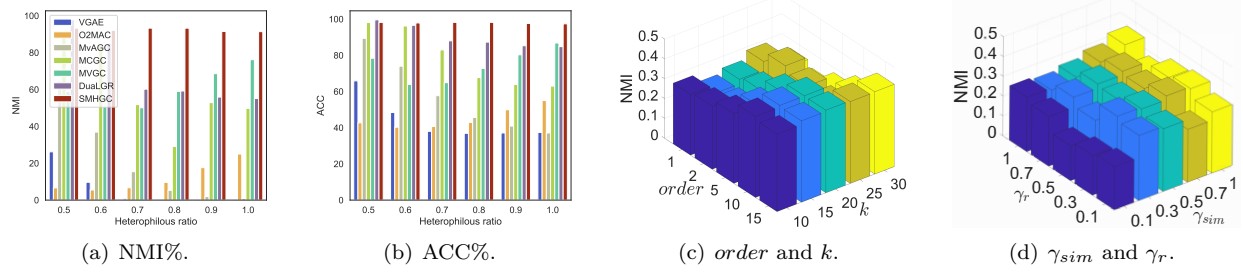

(a) NMI%.     (b) ACC%.     (c) *order* and *k*.     (d) $\gamma_{sim}$ and $\gamma_r$.

Figure 4: Clustering results on six semi-synthetic ACM datasets with different heterophilous ratios (Fig. 4(a) and Fig. 4(b)), and parameter sensitive analysis *w.r.t.* *order*, *k* (Fig. 4(c)), $\gamma_{sim}$ and $\gamma_r$ (Fig. 4(d)). The whole results can be found in Appendix.

performances of methods of O2MAC and MCGC start to increase when the heterophilous ratio achieved 0.7. This phenomenon partially demonstrated Observation 2, which implies that the larger heterophilous ratio of original graph does not mean worse clustering results. For instance, when the heterophilous ratio reaches 1.0, nodes with the same label connect to nodes in all other clusters. If the number of edges is sufficiently large, these nodes will have similar neighbor patterns, connecting to all nodes in other clusters. Thus, the increase in clustering performance with a higher heterophilous ratio may be due to the increased similarity of these neighbor patterns. Overall, Fig. 4(a) and Fig. 4(b) show that the performance generally significantly decreases compared to when dealing with homophilous graphs. This suggests that heterophilous graphs pose a challenge to existing MVGC methods. In contrast, leveraging similarity, SMHGC maintains relatively stable performance despite the increasing heterophilous ratio. This stability can be attributed to the extraction of homophilous information coupled with similarities, as well as intra- and inter-view fusion based on global similarity and consensus, empirically demonstrating the robustness of similarity for MVHGC.

Table 3: The ablation study of SMHGC on ACM and Texas.

| Compenents | ACM (*hr* 0.82 & 0.64) | | | | Texas (*hr* 0.09 & 0.09) | | | |
|---|---|---|---|---|---|---|---|---|
| | NMI% | ARI% | ACC% | F1% | NMI% | ARI% | ACC% | F1% |
| w/o $\mathcal{L}_{sim}$ | 52.7 | 48.3 | 66.5 | 65.8 | 20.2 | 31.7 | 61.8 | 36.2 |
| w/o $\mathcal{L}_r$ | 48.1 | 46.9 | 72.0 | 71.7 | 38.2 | 41.0 | 67.8 | 49.1 |
| w/o $\mathcal{L}_{kl}$ | 74.9 | 76.3 | 91.4 | 91.4 | 40.5 | 46.7 | 71.0 | 49.0 |
| w/o $\mathbf{A}_x^v$ | 39.1 | 35.2 | 66.4 | 66.0 | 22.3 | 26.9 | 60.1 | 34.5 |
| w/o $\mathbf{A}_a^v$ | 70.9 | 70.2 | 88.1 | 87.0 | 37.6 | 41.1 | 68.9 | 47.7 |
| w/o $\omega_x, \omega_a$ | 81.0 | 82.8 | 93.5 | 93.3 | 38.7 | 46.7 | 69.9 | 49.2 |
| **SMHGC** | **81.1** | **83.2** | **93.9** | **93.9** | **41.8** | **46.9** | **71.3** | **49.8** |

### 4.3 Parameter Sensitive Analysis

SMHGC primarily relies on two key hyperparameters: $k$ and $order$, which determine the number of edges in $\mathbf{S}^v$ and the aggregation order of $\mathbf{S}^v$, respectively. To explore their specific effects on the model, we conduct a parameter sensitivity analysis of these two hyperparameters on ACM and Texas datasets. Part of the results is depicted in Fig. 4(c). As illustrated, our SMHGC has better performance when $k$ is in different ranges depending on the datasets. This observation suggests that the model aggregates an optimal amount of homophilous messages from the neighborhood when $k$ is appropriately chosen. Empirically, selecting $k$ within 8% to 12% of the number of nodes appears to yield better results. Additionally, as $order$ increases, SMHGC exhibits a trend of initially rising and then stabilizing, with the peak performance observed within the range of 2 to 10. Furthermore, Fig. 4(d) illustrates the effect of different values of $\gamma_{sim}$ and $\gamma_r$ on the final results. Overall, the varying weights of $\mathcal{L}_{sim}$ and $\mathcal{L}_r$ have a relatively moderate impact on the model outcomes. Increasing the weight of $\mathcal{L}_{sim}$ slightly improves the final results. In this study, both $\gamma_{sim}$ and $\gamma_r$ are set to 1 according to the experimental results.

### 4.4 Ablation Study

**Effect of each loss.** As depicted in Table 3, the performance of SMHGC experiences a significant drop in the absence of $\mathcal{L}_{sim}$. This not only validates Proposition 1 empirically but also underscores the feasibility and effectiveness of exploring homophilous information in node features and neighbor patterns through the proposed similarity loss. Additionally, $\mathcal{L}_r$ represents the reconstruction loss of the node features, aiming to retain as much key information as possible. Its absence leads to a degradation in the model's performance across all metrics. On the other hand, $\mathcal{L}_{kl}$ contributes to obtaining distinguishable embeddings, although its impact appears to be subtle.

**Effect of couple similarities.** As observed in the third and fourth rows of Table 3, the absence of either $\mathbf{A}_x^v$ or $\mathbf{A}_a^v$ results in a certain degree of performance degradation. This indicates that both node features and neighbor patterns contain certain complementary homophilous information. Consequently, it underscores the necessity of mining homophilous information from node features and neighbor patterns, respectively.

**Effect of global similarity.** The absence of $\omega_x$ and $\omega_a$ have a negligible effect on the model's performance on ACM and a relatively strong effect on Texas. This discrepancy may arise from the varying relevance of node features and neighbor patterns to the downstream task across different datasets.

### 4.5 Analysis on Similarity Matrices

As shown in Fig. 5 and Fig. 6, the heatmaps highlight the contributions of different modules in capturing homophilous information. On ACM dataset, the raw feature similarity matrix $\mathbf{X}\mathbf{X}^{\mathrm{T}}$ reveals the basic relationships between node features but lacks distinct cluster patterns. After encoding, the feature similarity

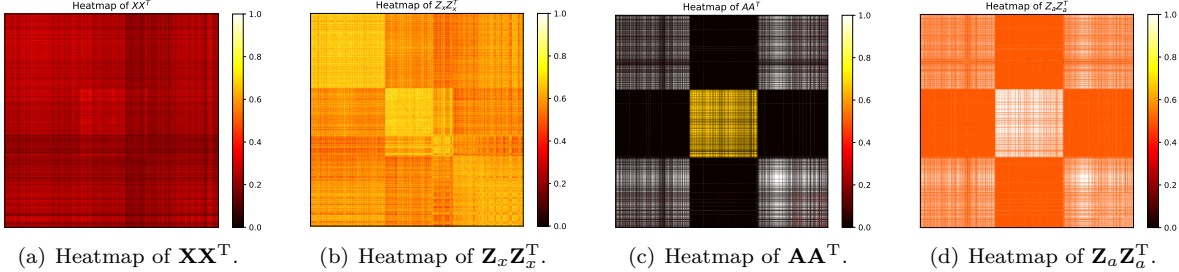

(a) Heatmap of $\mathbf{X}\mathbf{X}^{\mathrm{T}}$.
(b) Heatmap of $\mathbf{Z}_x\mathbf{Z}_x^{\mathrm{T}}$.
(c) Heatmap of $\mathbf{A}\mathbf{A}^{\mathrm{T}}$.
(d) Heatmap of $\mathbf{Z}_a\mathbf{Z}_a^{\mathrm{T}}$.

Figure 5: Heatmaps comparing raw and encoded similarity matrices on ACM ($\omega_x = 1, \omega_a = 0.12$).

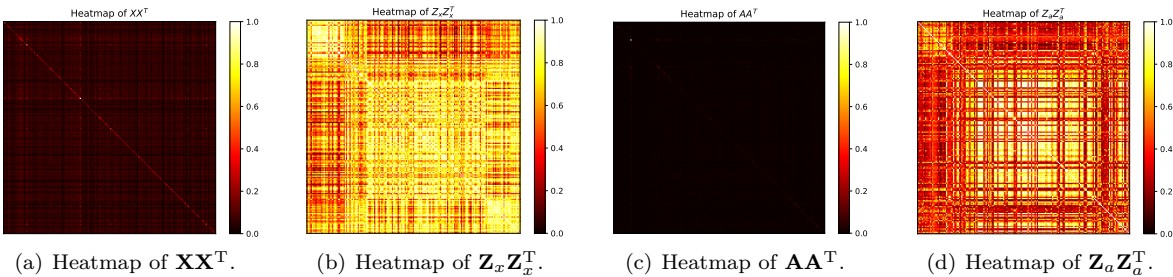

(a) Heatmap of $\mathbf{X}\mathbf{X}^{\mathrm{T}}$.
(b) Heatmap of $\mathbf{Z}_x\mathbf{Z}_x^{\mathrm{T}}$.
(c) Heatmap of $\mathbf{A}\mathbf{A}^{\mathrm{T}}$.
(d) Heatmap of $\mathbf{Z}_a\mathbf{Z}_a^{\mathrm{T}}$.

Figure 6: Heatmaps comparing raw and encoded similarity matrices on Texas ($\omega_x = 1, \omega_a = 0.26$).

matrix $\mathbf{Z}_x\mathbf{Z}_x^{\mathrm{T}}$ significantly enhances homophilous information by learning more precise feature representations, resulting in a clear block structure. Similarly, structural similarity, represented by the neighbor pattern similarity matrix $\mathbf{A}\mathbf{A}^{\mathrm{T}}$, reflects certain local structural patterns but remains scattered due to the interference of heterophilous edges. The encoded structural similarity matrix $\mathbf{Z}_a\mathbf{Z}_a^{\mathrm{T}}$ markedly improves the clarity of these patterns, extracting implicit homophilous information while effectively mitigating heterophilous noise. In the Texas dataset shown in Fig. 6, the raw feature similarity matrix struggles to capture meaningful patterns due to data sparsity, but the encoded feature similarity matrix still manages to reveal relatively clear homophilous structures. Likewise, structural similarity is more limited under sparsity, yet the encoded structural similarity matrix demonstrates enhanced discriminatory power. These results indicate that the joint optimization of feature and neighbor pattern similarities allows the model to extract effective homophilous information from both the feature and structural spaces, thereby improving clustering performance. Moreover, the results validate the complementarity between feature similarity and neighbor pattern similarity, as well as the critical role of the encoder in extracting meaningful information.

In addition, compare the learned intra-view weight pairs $\omega_x$ and $\omega_a$ (introduced in Section 3.3), we can observe that $\omega_x$ is larger than $\omega_a$, indicating the model emphasizes the feature similarity, which aligns with the visualizations that the heatmaps of $\mathbf{Z}_x\mathbf{Z}_x^{\mathrm{T}}$ (Figs. 5(b) and 6(b)) have better homophilous blocks than the heatmaps of $\mathbf{Z}_a\mathbf{Z}_a^{\mathrm{T}}$ ((Figs. 5(d) and 6(d))). Furthermore, this conclusion also aligns well with the conclusion from Table. 3, where component 'w/o $\mathbf{A}_x^v$' has the greatest influence on ACM and Texas datasets.

## 5 Conclusion

In this study,we analyzed the observation about homophily and similarity, and introduce an effective solution for multi-view heterophilous graph clustering, called SiMilarity-enhanced homophily for Multi-view Heterophilous Graph Clustering (SMHGC). Confronted with the challenges posed by heterophilous graphs, we empirically demonstrated the robust power of similarity for unsupervised clustering tasks. Our analysis explores how the similarity could enhance homophilous and clustering performance. Constructed on this foundation, we propose two regularization losses, *i.e.*, neighbor pattern similarity and node feature similarity, to enhance graph homophily under the guidance of introduced multi-view global similarity. Further, we

propose a paradigm for fusing inter- and intra-view information, enabling the integration of homophilous information from different sources and levels through the utilization of global similarity and multi-view consensus. Extensive experiments demonstrate the strong robustness of SMHGC for multi-view heterophilous graph clustering, validating the feasibility and effectiveness of our proposed solution.

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

# A   RELATED WORKS

## A.1   Multi-View Graph Clustering

With the advancement of GNNs, researchers are eager to explore the graph structural information for multi-view clustering. In recent years, an abundance of methods for MVGC have emerged. O2MAC (Fan et al., 2020) pioneered the application of GNNs in MVGC. Their approach is to encode multi-view graphs into a lower-dimensional space using a single-view graph convolutional encoder and a multi-view graph structure decoder. Cheng et al. (2021) proposed two-pathway graph encoders, which facilitate the mapping of graph embedding features and the acquisition of view-consistent information. Hassani & Khasahmadi (2020) proposed an innovative GNN-based solution aimed at acquiring node and graph level representations specifically for multi-view self-supervised learning. Pan & Kang (2021) leveraged the technique of contrastive learning to excavate the shared geometric and semantic patterns, thereby facilitating the learning of a consensus graph. MVGC (Xia et al., 2022b) systematically explored the cluster structure by employing a graph convolutional encoder, which is trained to learn the self-expression coefficient matrix. However, these methods demonstrate a high sensitivity to the homophily of graphs. Furthermore, DuaLGR, proposed by Ling et al. (2023), initially attempted to address the MVHGC problem. Despite its promising performance on heterophilous graphs, it struggles to explain the rationale of its performance enhancement, which limits its interpretability and generalizability. In contrast, our approach explores the relationship between similarity and homophilous information, providing reliable support for the MVHGC problem from a data-centric view.

## A.2   Heterophilous Graph Learning

Many works concentrate on single-view heterophilous graph learning. For instance, Li et al. (2022b) proposed a graph restructuring method that enhances spectral clustering by aligning with node labels. A feature extraction technique adaptable to both homophilous and heterophilous graphs devised by Chanpuriya & Musco (2022), demonstrating its effectiveness in node classification. These methods reduce the impact of heterophily but are challenging to apply to MVHGC due to their reliance on labels in supervised tasks. Additionally, unsupervised graph learning methods, such as GREET (Liu et al., 2023), which uses an edge discriminator to differentiate between homophilous and heterophilous edges, and Xiao et al. (2022) developed the decoupled self-supervised learning (DSSL) framework. While these approaches can handle heterophilous graphs, they are complex and often designed for single-view tasks, making them difficult to generalize to multi-view settings. Effective solutions for multi-view heterophilous graphs remain limited.

# B   Generalization Analysis

To assess the robustness and generalizability of the proposed similarity terms, we conduct a generalization analysis. As a way to infer the generalization bound of the similarity, we introduce the hypothesis function $h_a \colon \mathcal{A} \to \mathbb{R}^{d_a}$ and $h_x \colon \mathcal{X} \to \mathbb{R}^{d_x}$ to map the neighbor pattern and node feature into the embedding points, where $\mathcal{A}$ and $\mathcal{X}$ represent the neighbor pattern and node feature space respectively. Then the embedding points from the neighbor pattern and node feature can be obtained by $z_{a_i} := h_a(a_i)$ and $z_{x_i} := h_x(x_i)$. Suppose $\mathcal{H}$ represents the family of $h$. On the basis of this, the similarity loss can be expressed as:

$$
\begin{aligned}
\mathcal{L}_{sim_a} &= \|\mathbf{Z}_a \mathbf{Z}_a^{\mathrm{T}} - \mathbf{A}\mathbf{A}^{\mathrm{T}}\|^2 = \frac{1}{N} \sum_{i=1}^{N} \sum_{j=1}^{N} \|h_a(a_i) h_a(a_j)^{\mathrm{T}} - a_i a_j^{\mathrm{T}}\|^2, \\
\mathcal{L}_{sim_x} &= \|\mathbf{Z}_x \mathbf{Z}_x^{\mathrm{T}} - \mathbf{X}\mathbf{X}^{\mathrm{T}}\|^2 = \frac{1}{N} \sum_{i=1}^{N} \sum_{j=1}^{N} \|h_x(x_i) h_x(x_j)^{\mathrm{T}} - x_i x_j^{\mathrm{T}}\|^2.
\end{aligned}
\tag{14}
$$

For the similarity loss from neighbor patterns $\mathcal{L}_{sim_a}$, the empirical risk can be defined as:

$$
\hat{\mathcal{L}}_a(h) = \frac{1}{N} \sum_{i=1}^{N} \sum_{j=1}^{N} \|h_a(a_i) h_a(a_j)^{\mathrm{T}} - a_i a_j^{\mathrm{T}}\|^2.
\tag{15}
$$

**Theorem 1.** *Let $\mathcal{L}(h)$ be the expectation of $\hat{\mathcal{L}}(h)$. Suppose that for any $a \in \mathcal{A}$ and $h \in \mathcal{H}$, there exists $M < \infty$ such that $\|aa^{\mathrm{T}}\|$, $\|h_a(a)h_a(a)^{\mathrm{T}}\| \in [0, M]$ hold. Then with probability $1 - \tau$ for any $h \in \mathcal{H}$ the inequality holds:*

$$\hat{\mathcal{L}}(h) \leq \mathcal{L}(h) + 2\sqrt{2}M^2\sqrt{N}(4 + 3\sqrt{\log\frac{1}{\tau}}). \tag{16}$$

Using Theorem 1, it can be easily found that the expected risk of the similarity is bounded by the empirical risk on input neighbor patterns and node features and the complexity term that depends on $M$ and $N$. The proof and model complexity analysis are presented in the Appendix.

**Theorem 2.** *Let $\mathcal{L}(h)$ be the expectation of $\hat{\mathcal{L}}(h)$. Suppose that for any $a \in \mathcal{A}$ and $h \in \mathcal{H}$, there exists $M < \infty$ such that $\|aa^{\mathrm{T}}\|$, $\|h_a(a)h_a(a)^{\mathrm{T}}\| \in [0, M]$ hold. Then with probability $1 - \tau$ for any $h \in \mathcal{H}$ the inequality holds:*

$$\hat{\mathcal{L}}(h) \leq \mathcal{L}(h) + 2\sqrt{2}M^2\sqrt{N}(4 + 3\sqrt{\log\frac{1}{\tau}}). \tag{17}$$

### B.1   Proof of Theorem 1

*Proof.* For the given neighbor pattern set $P = \{a_1, a_2, \cdots, a_N\}$, let $P'$ be the neighbor pattern set where only one neighbor pattern $\bar{a}_r$ differs from $P$, and let $\hat{\mathcal{L}}'(h)$ represent the empirical risk of $h$ on $P'$. We can have:

$$
\begin{aligned}
&|\sup_{h \in \mathcal{H}} |\hat{\mathcal{L}}(h) - \mathcal{L}(h)| - \sup_{h \in \mathcal{H}} |\hat{\mathcal{L}}'(h) - \mathcal{L}(h)||\\
&\leq \sup_{h \in \mathcal{H}} |\hat{\mathcal{L}}(h) - \hat{\mathcal{L}}'(h)|\\
&= \sup_{h \in \mathcal{H}} |\frac{2}{N}\sum_{j=1}^{N} \|h_a(a_r)h_a(a_j)^{\mathrm{T}} - a_r a_j^{\mathrm{T}}\|^2\\
&\quad - \|h_a(\bar{a}_r)h_a(a_j)^{\mathrm{T}} - \bar{a}_r a_j^{\mathrm{T}}\|^2|\\
&= \sup_{h \in \mathcal{H}} \frac{2}{N}|\sum_{j=1}^{N} \|h_a(a_r)h_a(a_j)^{\mathrm{T}}\|^2\\
&\quad - \|h_a(\bar{a}_r)h_a(a_j)^{\mathrm{T}}\|^2 + \|a_r a_j^{\mathrm{T}}\|^2 - \|\bar{a}_r a_j^{\mathrm{T}}\|^2\\
&\quad + 2\|h_a(a_r)h_a(a_j)^{\mathrm{T}}\|\|a_r a_j^{\mathrm{T}}\|\\
&\quad + 2\|h_a(\bar{a}_r)h_a(a_j)^{\mathrm{T}}\|\|\bar{a}_r a_j^{\mathrm{T}}\||\\
&\leq 12M^2.
\end{aligned}
\tag{18}
$$

In order to bound the expectation term $\mathbb{E}_P \sup_{h \in \mathcal{H}} |\mathcal{L}(h) - \hat{\mathcal{L}}(h)|$, let $\sigma_1, \sigma_2, \cdots, \sigma_N$ be *i.i.d.* Rademacher random variables, we can have:

$$
\begin{aligned}
&\mathbb{E}_P \sup_{h \in \mathcal{H}} |\mathcal{L}(h) - \hat{\mathcal{L}}(h)|\\
&\leq \mathbb{E}_{P,P} \sup_{h \in \mathcal{H}} |\frac{1}{N}\sum_{i=1}^{N}\sum_{j=1}^{N} \|h_a(a_i)h_a(a_j)^{\mathrm{T}}\\
&\quad - a_i a_j^{\mathrm{T}}\|^2 - \|h_a(\hat{a}_i)h_a(\hat{a}_j)^{\mathrm{T}} - \hat{a}_i \hat{a}_j^{\mathrm{T}}\|^2|\\
&= 2\mathbb{E}_{P,\sigma} \sup_{h \in \mathcal{H}} |\frac{2}{N}\sum_{i=1}^{N}\sigma_i \sum_{j=1}^{N} \|h_a(a_r)h_a(a_j)^{\mathrm{T}} - a_r a_j^{\mathrm{T}}\|^2|.
\end{aligned}
\tag{19}
$$

According to Khintchine-Kahane inequality, Eq. (19) can be bounded as:

$$
\begin{aligned}
2\mathbb{E}_{P,\sigma} &\sup_{h\in\mathcal{H}} |\frac{2}{N}\sum_{i=1}^{N}\sigma_i\sum_{j=1}^{N}\|h_a(a_r)h_a(a_j)^{\mathrm{T}} - a_r a_j^{\mathrm{T}}\|^2| \\
&\leq 2\mathbb{E}_{P,\sigma}\sup_{h\in\mathcal{H}}(\frac{2}{N}\sum_{i=1}^{N}(\sum_{j=1}^{N}\|h_a(a_r)h_a(a_j)^{\mathrm{T}} - a_r a_j^{\mathrm{T}}\|^2)^2)^{\frac{1}{2}} \\
&\leq 8\sqrt{2}M^2\sqrt{N}.
\end{aligned}
\tag{20}
$$

Substituting Eq. (20) into Eq. (19), we have:

$$
\mathbb{E}_P \sup_{h\in\mathcal{H}} |\mathcal{L}(h) - \hat{\mathcal{L}}(h)| \leq 8\sqrt{2}M^2\sqrt{N}.
\tag{21}
$$

Finally, according to the McDiarmid inequality, it can be conclued that with probability $1-\tau$ for any $h\in\mathcal{H}$ the following inequality holds:

$$
\begin{aligned}
\hat{\mathcal{L}}(h) &\leq \mathcal{L}(h) + 8\sqrt{2}M^2\sqrt{N} + 6\sqrt{2}M^2\sqrt{\log\frac{1}{\tau}}\sqrt{N} \\
&= \mathcal{L}(h) + 2\sqrt{2}M^2\sqrt{N}(4 + 3\sqrt{\log\frac{1}{\tau}}).
\end{aligned}
\tag{22}
$$

For the similarity loss from node features $\mathcal{L}_{sim_x}$, the proof and derivation procedure is the same as above. □

## C Complexity Analysis

Considering $N$, $K$, $V$, $d$ and $order$ as the number of nodes, the number of cluster centroids, the number of views, the maximum dimensionality of input $\mathbf{X}^v$ and the order of aggregation. Let $L$ denote the maximum number of neurons in MLP's hidden layers (the encoders and decoders are instantiated with MLPs in this work), and $d_h$ as the dimensionality of the embedding ($\mathbf{H}^v$, $\overline{\mathbf{H}}$). The complexity of the encoders and decoders ($f_a^v$, $f_x^v$, $f_{\phi^v}^v$ and $g_{\xi^v}^v$) from the V views is $O(VNL^2)$. The similarity calculated in Eq. (5) needs $O(VN^2)$. The complexity of the aggregation process in Eq. (10) is $O(VdN^2 order)$, which is depend on the aggregation order $order$. In addition, the evaluation function (instantiated with $K$-means) in Eq. (11) needs $O(VNKd_h)$. In summary, the complexity of SMHGC is $O(VN(L^2 + Kd_h) + dVN^2 order)$, which is proportional to the square of the number of nodes $N^2$ and the aggregation order $order$.

Table 4: The statistics information of the four graph datasets. $hr$ is the homophilous ratio, which describes the ratio of #Homo-edges and #Edges.

| Datasets | #Clusters | #Nodes | #Features | #Graphs | #Homo-edges | #Edges | $hr$ |
|---|---|---|---|---|---|---|---|
| ACM | 3 | 3025 | 1870 | $\mathcal{G}^1$ | 21550 | 26252 | 0.82 |
| | | | 1870 | $\mathcal{G}^2$ | 1411658 | 2207736 | 0.64 |
| DBLP | 4 | 4057 | 334 | $\mathcal{G}^1$ | 5636 | 7056 | 0.80 |
| | | | 334 | $\mathcal{G}^2$ | 3346042 | 4996438 | 0.67 |
| | | | 334 | $\mathcal{G}^3$ | 2183134 | 6772278 | 0.32 |
| Texas | 5 | 183 | 1703 | $\mathcal{G}^1$ | 50 | 574 | 0.09 |
| | | | 1703 | $\mathcal{G}^2$ | 50 | 574 | 0.09 |
| Chameleon | 5 | 2277 | 2325 | $\mathcal{G}^1$ | 14476 | 62792 | 0.23 |
| | | | 2325 | $\mathcal{G}^2$ | 14476 | 62792 | 0.23 |

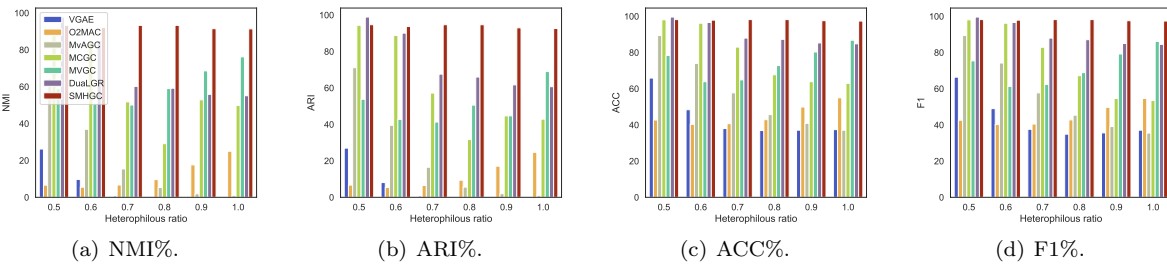

(a) NMI%.      (b) ARI%.      (c) ACC%.      (d) F1%.

Figure 7: Clustering results on six semi-synthetic ACM datasets with different heterophilous ratios.

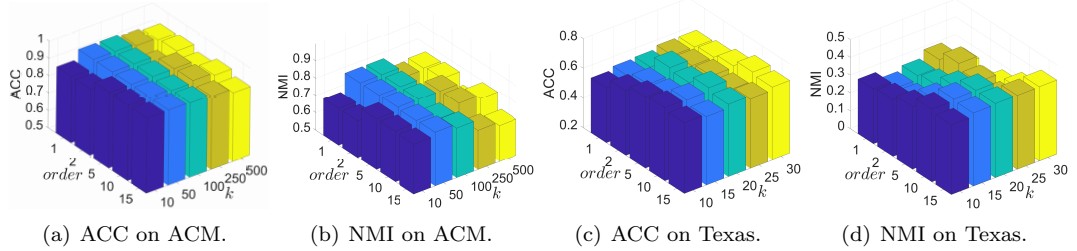

(a) ACC on ACM.      (b) NMI on ACM.      (c) ACC on Texas.      (d) NMI on Texas.

Figure 8: Sensitive analysis of ACC and NMI on ACM and Texas with *order* and $k$.

# D  More details of SMHGC

## D.1  Kullback-Leibler divergence loss

In this multi-view clustering task, we apply Kullback-Leibler divergence loss ($\mathcal{L}_{kl}$) to facilitate the model to obtain distinguishable embedding following the previous related work Ren et al. (2024); Xie et al. (2016b). Specifically, let $q_{ij}^v \in \mathbf{Q}^v$ be the soft cluster assignment that describes the possibility of node $i$ belongs to cluster centroid $j$ in $v$-th view and $\mathbf{P}$ be the sharpen target distribution, the final loss can be expressed as:

$$\mathcal{L}_{kl} = \mathrm{KL}(\overline{\mathbf{P}}\|\overline{\mathbf{Q}}) + \sum_{v=1}^{V} \mathrm{KL}(\overline{\mathbf{P}}\|\mathbf{Q}^v), \tag{23}$$

where $\overline{\mathbf{Q}}$ and $\overline{\mathbf{P}}$ represent the soft and target distribution of $\overline{\mathbf{H}}$, respectively. Following Xie et al. (2016b), the soft assignment $q_{ij}$ and sharpen assignment $p_{ij}$ can be expressed by the utilization of Student's $t$-distribution as a kernel:

$$\begin{aligned} q_{ij} &= \frac{(1 + \|h_i - \mu_j\|^2/\alpha)^{-\frac{\alpha+1}{2}}}{\sum_{j'} 1 + \|h_i - \mu_{j'}\|^2/\alpha)^{-\frac{\alpha+1}{2}}}, \\ p_{ij} &= \frac{q_{ij}^2/\sum_i q_{ij}}{\sum_{j'} (q_{ij'}^2/\sum_i q_{ij'})}, \end{aligned} \tag{24}$$

where $h_i$ denotes the final embedding, $\mu_j$ is the centroid embedding of $j$-th cluster, and $\alpha$ denotes the degrees of freedom of the Student's $t$-distribution which is setting to 1 for all experiments.

With Eq. (23), we expect the first term to enhance the discriminability of the final embedding $\overline{\mathbf{H}}$, followed by the second term to encourage the soft distribution of each view to fit the distribution of $\overline{\mathbf{H}}$, and thus to exploit the inter-view consistency.

## D.2  Mathematical Definition and Discussion about Hypothesis 1

An *ideal subspace* $\mathcal{Z}^*$ for node clustering is a lower-dimensional subspace of the original feature space $\mathcal{X}$ that maximizes the intra-cluster similarity while minimizing the inter-cluster similarity. Formally, let $\mathbf{X} \in \mathbb{R}^{n \times d}$

be the matrix of node embeddings, where $n$ is the number of nodes and $d$ is the dimensionality of the feature space. The ideal subspace $\mathcal{Z}^*$ is defined as:

$$\mathcal{Z}^* = \arg\max_{\mathcal{Z} \subseteq \mathcal{X}} \mathcal{J}(\mathbf{X}_{\mathcal{Z}})$$

where $\mathbf{X}_{\mathcal{Z}}$ is the projection of $\mathcal{X}$ onto the subspace $\mathcal{Z}$, and $\mathcal{J}$ is an objective function that quantifies the quality of the subspace. The objective function $\mathcal{J}$ can be defined in several ways, but in the context of MVHGC, the choice is to maximize the intra-cluster similarity and minimize the inter-cluster similarity. For instance, one possible formulation is:

$$\mathcal{J}(\mathbf{X}_{\mathcal{Z}}) = \frac{\sum_{i,j \in C_k} \text{sim}(\mathbf{x}_{i\mathcal{Z}}, \mathbf{x}_{j\mathcal{Z}})}{\sum_{i \in C_k, j \in C_l, k \neq l} \text{sim}(\mathbf{x}_{i\mathcal{Z}}, \mathbf{x}_{j\mathcal{Z}})},$$

where $\text{sim}(\cdot, \cdot)$ is a similarity measure (e.g., cosine similarity) between the projections of nodes $i$ and $j$ in the subspace $\mathcal{Z}$. $C_k$ and $C_l$ are clusters in the clustering result. Intuitively, the defined ideal subspace should have the following properties:

- **Intra-Cluster Similarity:** Nodes within the same cluster should have high similarity in the ideal subspace. This ensures that nodes with similar features are clustered together.

- **Inter-Cluster Dissimilarity:** Nodes from different clusters should have low similarity in the ideal subspace. This ensures that nodes with dissimilar features are separated into different clusters.

In practical, SMHGC tries to learn such ideal subspace in the autoencoder-like architecture, and regularized by *neighbor pattern similarity* and *feature similarity* terms ($\mathcal{L}_{sim}$), and the multi-view collaborative clustering loss $\mathcal{L}_{KL}$.

**An example of ideal subspace.** Consider a simple example where we have a graph with nodes representing documents and edges representing co-citations. The original feature space might include bag-of-words representations of the documents. An ideal subspace could be a lower-dimensional space where documents from the same topic are close to each other, and documents from different topics are far apart.

## E    Details of Experiments

### E.1    Datasets

Both homophilous and heterophilous graph datasets are considered in this work to evaluate our model. Besides, six datasets were synthesized to obtain a smoother and more intuitive evaluation regarding to homophily ratio ($hr$). The statistics information of these datasets is summarized in Table 4. Particularly, ACM [4] and DBLP [5] are two widely used homophilous multi-view graph datases, which are two citation networks that contain two and three graph respectively. Texas and Chameleon are two widely used heterophilous grah datasets, which are a webpage graph from WebKB [6] and a subset of the Wikipedia network Rozemberczki et al. (2021). As Texas and Chameleon are single view graph data, we duplicate the feature and graph as a second view. In addition, we take ACM as an example to generate different $hr$ graphs with random sampling, each view's graph with the same number of edges as the original view for evaluation, whose $hr$ range from 0.0 to 0.5 with the step of 0.1 Ling et al. (2023).

### E.2    Implementation Details

---

[4]https://dl.acm.org/
[5]https://dblp.uni-trier.de/
[6]http://www.cs.cmu.edu/afs/cs.cmu.edu/project/theo-11/www/wwkb

### E.2.1   Implementation Details on Figure 1 (a)

In Figure 1(a), we explored the inherent Limitation of GNNs for Clustering on Heterophilous Graph. The experiments follow the following settings, expecting to result a robust Observation.

- **Model selection**: To exclude the influential factors from model design, optimization influence, we design a very basic GNN that contains only two non-parametric message passing layers. The basic GNN can be formulated as below. After $\mathbf{h}^2$ obtained, we conduct K-means to get final clustering results and compute its NMI scores.

$$\mathbf{h}_i^1 = \mathbf{x}_i + \sum_{j \in \mathcal{N}(i)} \mathbf{x}_j, \mathbf{h}_i^2 = \mathbf{h}_i^1 + \sum_{j \in \mathcal{N}(i)} \mathbf{h}_j^1$$

- **Data selection**: We choose the two most widely used graph datasets Cora and Citeseer, and refine the two graphs to obtain different homophilous ratio. Specifically, to control their homophilous ratio, we keep their edge number to be unchanged, and reorganize their edges by randomly dropping homophilous edges and adding new homophilous edges.

### E.3   Implementation Details of HR of Neighbor Pattern Similarity

### E.3.1   Formal Definition of HR of Neighbor Pattern Similarity

Following previous works, HR (Homophilous Ratio) is defined as the proportion of edges that connect nodes in a same cluster:

$$\text{HR} = \frac{1}{|\mathcal{E}|} \sum_{(i,j) \in \mathcal{E}} \delta(\mathbf{x}_i, \mathbf{x}_j),$$

where $\mathcal{E}$ is the set of edges in the graph, $\mathbf{x}_i$ and $\mathbf{x}_j$ are the feature vectors of nodes $i$ and $j$, and $\delta(\mathbf{x}_i, \mathbf{x}_j)$ is an indicator function that equals 1 if $\mathbf{x}_i$ and $\mathbf{x}_j$ are similar (e.g., based on a predefined similarity threshold) and 0 otherwise.

In the context of neighbor pattern similarity, the computation of HR conducts on neighbor pattern similarity matrix $\mathbf{A}\mathbf{A}^{\mathrm{T}}$, $\mathbf{x}_i$ and $\mathbf{x}_j$ refer the set of neighbors of node $i$ and node $j$. Specifically, let $\mathbf{N}_i$ denote the set of neighbors of node $i$, and let $c_i$ be the class label of node $i$. The neighbor pattern similarity between nodes $i$ and $j$ can be measured by the similarity of their neighbor sets $\mathbf{N}_i$ and $\mathbf{N}_j$.

The HR of neighbor pattern similarity can be defined as:

$$\text{HR} = \frac{1}{|\mathcal{E}|} \sum_{(i,j) \in \mathcal{E}} \delta(\mathbf{N}_i, \mathbf{N}_j),$$

where $\delta(\mathbf{N}_i, \mathbf{N}_j)$ is an indicator function that equals 1 if the neighbor sets $\mathbf{N}_i$ and $\mathbf{N}_j$ are similar (e.g., based on a predefined similarity threshold 0.5) and 0 otherwise.

### E.3.2   Increasing HR of Neighbor Pattern Similarity

To increase the HR of neighbor pattern similarity, we leverage the insight from Proposition 1, which suggests that "good heterophily" corresponds to similar neighbor pattern distributions. Our approach involves modeling the neighbor distribution of each class as a Gaussian distribution and reducing the standard deviation (std) of these distributions. This reduction in std enhances the HR of neighbor pattern similarity. Here is a two steps description of the method: **Step1: Modeling Neighbor Distributions:** Assume that the neighbor distribution of nodes within each class follows a Gaussian distribution. Let $\mathcal{N}(\mu_c, \sigma_c^2)$ represent the Gaussian distribution for class $c$, where $\mu_c$ is the mean and $\sigma_c$ is the std. **Step2: Reducing Standard Deviation:** By reducing the standard deviation $\sigma_c$, we make the neighbor distributions more concentrated around their means. This concentration increases the likelihood that nodes within the same class will have similar neighbor patterns, thereby enhancing the homophilous ratio (HR).

To explain how the std influence HR, we give the following examples:

- **High Standard Deviation ($\sigma_c$ is large):** When the std is large, the neighbor distribution becomes more spread out, leading to a more uniform distribution. In this case, nodes are likely to connect to other nodes randomly, resulting in a lower HR.

- **Low Standard Deviation ($\sigma_c$ is small):** When the std is small, the neighbor distribution becomes more concentrated. Nodes within the same class are more likely to connect to a specific subset of nodes, leading to higher similarity in neighbor patterns and a higher HR.

By reducing the std of the neighbor distributions, we can effectively increase the HR of neighbor pattern similarity as shown in Figure 3.

## F    More Experimental Results

### F.1    Complete Clustering Results about Clustering Results on Six Semi-synthetic ACM Datasets

The whole results about clustering results on six semi-synthetic ACM datasets with different heterophilous ratios are shown in Fig. 7. These results suggest that heterophilous graphs are a barrier to the existing MVGC methods, while our SMHGC still has a relatively smooth performance with the power of similarity.

### F.2    Complete Clustering Results about Parameter Sensitive Analysis

The whole results about the parameter sensitive analysis of two hyperparameters, *i.e. k* and *order*, on ACM and Texas are shown in Fig. 8. Specifically, SMHGC seems to have a stable performance in terms of ACC on both ACM and Texas, where ACC on ACM is mainly between 0.85 and 0.95 and on Texas is mainly between 0.55 and 0.75. The NMI, on the other hand, is relatively more volatile, with the NMI on ACM fluctuating mainly between 0.65 and 0.80, and on Texas lying mainly between 0.25 and 0.40.

## G    Hyperparameters

Table 5: The hyperparameters used in the experiments.

| Parameter | ACM | Texas | Chameleon | DBLP |
|---|---|---|---|---|
| *order* | 2 | 10 | 10 | 6 |
| $\rho_h$ | 0.5 | 3 | 3 | 3 |
| $\rho_S$ | 0.5 | 3 | 3 | 3 |
| $k$ | 300 | 30 | 250 | 300 |
| Hidden dimension of $f_x$ | 128 | 128 | 1024 | 128 |
| Output dimension of $f_x$ | 16 | 32 | 32 | 64 |
| Hidden dimension of $f_a$ | 128 | 512 | 1024 | 256 |
| Output dimension of $f_a$ | 16 | 128 | 32 | 128 |
| Hidden dimension of $f_{\phi^v}$ | 128 | 128 | 1024 | 128 |
| Output dimension of $f_{\phi^v}$ | 16 | 32 | 32 | 64 |
| Number of layers $f_x$ | 2 | 4 | 4 | 2 |
| Number of layers $f_x$ | 2 | 4 | 4 | 4 |
| Learning rate | 0.0005 | 0.001 | 0.005 | 0.005 |
| Weight_decay | 0.0 | 0.0001 | 0.005 | 5e-7 |

Table 5 summarizes the hyperparameters used for the datasets (ACM, Texas, Chameleon, and DBLP) in the experiments. The values are tailored for each dataset to achieve optimal performance, ensuring fair and consistent evaluation across different graph datasets.

