# OpenReview forum: "Similarity-Enhanced Homophily for Multi-View Heterophilous Graph Clustering"
_TMLR — Rejected by TMLR_

### Review · Reviewer_suHe · 2024-11-13

**Summary Of Contributions:**

This paper identifies the limitations of current message-passing-based clustering methods in multi-view heterophilous graphs, where they are prone to degraded performance. To address this, the authors propose the "Similarity-Enhanced Homophily" framework (SMHGC), which leverages similarity measures—neighbor pattern similarity, node feature similarity, and multi-view global similarity—to improve clustering performance in an unsupervised manner. Empirical evaluations across four datasets demonstrate the effectiveness of SMHGC, highlighting its robustness for both homophilous and heterophilous graphs.

**Audience:**

Yes

**Claims And Evidence:**

No

**Requested Changes:**

If possible, please address my concerns above.

**Strengths And Weaknesses:**

## Pros
1. The paper is well written, with a clear-structured presentation of the problem, methodology, and experimental results. The authors define complex concepts like heterophily and homophily with clarity.
2. The paper logically flows from problem identification to proposed solutions and experimentation.
3. The authors provide extensive experimental evaluations, including comparisons with state-of-the-art methods on both homophilous and heterophilous datasets, as well as an ablation study.

## Cons
1. The conclusions about the limitations of GNNs in heterophilous graphs are not sufficiently supported by the single observation presented in _Observation 1_, where a basic model performs over two single-view graphs (without more specific details about this experiment). This is a weak argument to support the need for a new framework.
2. Many designs in SMHGC rely on simplistic assumptions, such as the neighbor pattern similarity can capture homophilous information. The lack of in-depth theoretical justification for these assumptions weakens the framework's credibility.
3. The details of the training process of SMHGC are not clear. The authors should provide more information about the training process, such as whether the model is trained end-to-end or needs a pre-training step for each view. Especially for the encoder-decoder pair of each view, it is crucial to determine the clustering performance of SMHGC. Additionally, the experimental settings should be described in more detail, including hyperparameters, learning rate, number of epochs, and so on.
4. To accommodate the discrete nature of the graph, the authors discretize the dense graph $S^{(v)}$. However, this process may impact the gradient flow during training and thus hinder the model's performance. The authors should provide more details on how they handle this issue and how to achieve the statement "Note that the conducted aggregation in Eq. (10) can be replaced by any complicated GNNs" considering the differentiability of the model.
5. The experimental section includes comparisons with only six multi-view clustering methods across four datasets, which is relatively limited and insufficient to fully evaluate SMHGC's performance.
6. The authors should provide a more comprehensive description of the synthetic datasets used in the experiments. In Figure 4b, where results for some methods increase with rising HR values, this trend seems counter-intuitive, as higher heterophily typically complicates clustering the authors stated. The authors should clarify this phenomenon.
7. The authors should consider providing a detailed analysis of the learned similarity weight pairs $\omega_x$ and $\omega_a$ across different datasets and HR levels. Since Table 2 indicates that the node feature similarity component $A^v_x$ has the greatest impact on SMHGC’s performance, visualizing the learned distributions could offer valuable insight into the respective contributions of node feature and neighbor pattern similarity.

---

> ### Author Response · Authors · 2024-12-06
> **Part1 Response**
>
> Thank you for your reviews. We response your comments question-by-question below, and the corresponding revisions conducted in the manuscript are colored in $\color{darkgreen}{darkgreen}$.
>
> ### 1. Insufficient Support for Limitations of GNNs in Heterophilous Graphs
>
> > The conclusions about the limitations of GNNs in heterophilous graphs are not sufficiently supported by the single observation presented in Observation 1, where a basic model performs over two single-view graphs (without more specific details about this experiment). This is a weak argument to support the need for a new framework.
>
> Thanks for your comments. To address your concern, we included the detailed experimental details for Observation 1.
>
> **Experimental details of Observation 1**
>
> *In Figure 1(a), we explored the inherent Limitation of GNNs for Clustering on Heterophilous Graph. The experiments follows the following settings, expecting to lead a robust Observation.*
>
> * **Model selection**: *to exclude the influential factors from model design, optimization influence, we design a very basic GNN that contains only two non-parametric message passing layers. The basic GNN can be formulated as below. After $\mathbf{h}^2$ obtained, we conduct K-means to get final clustering results and compute its NMI scores.*
>
> $$
> \mathbf{h}^1_i = \mathbf{x}_i + \sum{\mathbf{x}_j},
> $$
> $$
> \mathbf{h}^2_i = \mathbf{h}^1_i + \sum{\mathbf{h}^1_j}, \text{where} j \in \mathcal{N}(i)
> $$
> * **Data selection**: *we choose two most widely used graph dataset Cora and Citeseer, and refine the two graphs to obtain different homophilous ratio. Specifically, to control their homophilous ratio, we keep their edge number to be unchanged, and reorganize their edges by randomly dropping homophilous edges and adding new homophilous edges.*
>
>
>
> **Observation on real-world multi-view dataset DBLP**
>
> In addition, following the reviewers' suggestions to show that a similar observation exists in real-world multi-view datasets with more complex GNNs, we further conduct the two-layer message passing-based method Graph Auto Encoder (GAE, which is the most important baseline of graph clustering method)  on the three views of DBLP and the two views of ACM. We can observe that the results generally decrease with the increase of the heterophilous ratio. The values in brackets denote corresponding heterophilous ratio.
>
> **Clustering results on three views of DBLP and ACM:**
>
> |      | DBLP-view1（0.20） | DBLP-view2 (0.33) | DBLP-view3 (0.68) | ACM-view1 (0.82) | ACM-view2 (0.64) |
> | ---- | ------------------ | ----------------- | ----------------- | ---------------- | ---------------- |
> | ACC% | 66.06              | 76.53             | 41.36             | 50.28            | 67.83            |
> | NMI% | 41.23              | 63.43             | 10.65             | 26.53            | 45.06            |
> | ARI% | 37.16              | 59.80             | 8.41              | 13.51            | 40.25            |
> | F1%  | 55.85              | 64.53             | 30.80             | 52.46            | 68.66            |

---

> ### Author Response · Authors · 2024-12-06
> **Part2 Response**
>
> ### 2. Simplistic Assumptions in SMHGC Design
> > Many designs in SMHGC rely on simplistic assumptions, such as the neighbor pattern similarity can capture homophilous information. The lack of in-depth theoretical justification for these assumptions weakens the framework's credibility.
>
> We appreciate your comments regarding the SMHGC design. To address your concern, we clarified the assumptions included in SMHGC, and refined our manuscript for a more clear presentation.
>
> **Clarification for assumptions in SMHGC design: **
>
> First of all, we have to clarify that the assumption that neighbor pattern similarity must capture homophilous information is not essential. Alternatively, this is an observation or fact on real-world datasets rather than an essential assumption, as stated below.
>
> *Observation 2: On heterophilous graph (Texas and Chameleon), homophilous ratio of neighbor pattern similarity (Definition 4) and feature similarity could be better than the original adjacent.*
>
> This observation is a motivation that motivates the concept of pattern similarity as a regularization. However, in the design of SMHGC, this assumption is not a necessary condition for a good clustering performance. For example, there are several specific designs in SMHGC that are able to address the problem when the neighbor pattern similarity are still in a low homophilous ratio:
>
> * **Subspace learning:** In homophilous information extraction module, we do not use the neighbor pattern similarity $\mathbf{A}\mathbf{A}^\mathrm{T}$ directly, but consider this as a regularization term to inject neighbor pattern similarity information to the subspace. Moreover, the subspace is not only optimized by the neighbor pattern similarity, but also optimized by the loss computed with other views, thereby the learned subspace representations are not limited by the homophily of neighbor pattern similarity of this view.
> * **Multi-view collaborative training:** In the objective function, the parameters in every view are collaboratively trained by other views and multi-view consensus clustering performance. This provides a more comprehensive consideration in the subspace learning process, avoiding the reliance on the homophilous ratio of neighbor pattern similarity or feature similarity.
> * **Global similarity guided intra-view fusion:** After the homophilous information extraction module, the global similarity which contains multi-view consensus information is conducted as a proxy to evaluate the homophilous ratio of learned similarity matrices. This design could further avoid the issue when neighbor patterns cannot capture homophilous information.
> * **Global consensus-based inter-view fusion:** Moreover, if the final learned view-specific embedding is still in bad quality, the global consensus-based inter-view fusion could de-emphasize the weight of this view before obtaining final embedding.
>
> Thanks again for your suggestion to clarify this problem. We have simplified this discussion and included them in the revised manuscript.

---

> ### Author Response · Authors · 2024-12-06
> **Part3 Response**
>
> ### 3. Lack of Clarity in Training Process
> > The details of the training process of SMHGC are not clear. The authors should provide more information about the training process.
>
> We greatly appreciate your constructive suggestions on providing detailed training process. Below, we response to your specific question at first, and provide a more comprehensive description on our training process in the appendix of the revised manuscript.
>
> > Whether the model is trained end-to-end or needs a pre-training step for each view. Especially for the encoder-decoder pair of each view, it is crucial to determine the clustering performance of SMHGC.
>
> We appreciate your deep intuition and concern. The SMHGC is proposed as an end-to-end framework without any additional consideration of multi-step optimization. For the encoder-decoder pair, we've also considered pretrain an autoencoder before collaborative training, but which obtains a worse performance. Therefore, all parameters are randomly initialized and trained together. To address your concern, we show the clustering performance of pretraining an autoencoder  on ACM dataset.
>
> |                                     | NMI    | ACC    | ARI    | F1     |
> | ----------------------------------- | ------ | ------ | ------ | ------ |
> | Pretrained autoencoder              | 0.3222 | 0.6906 | 0.3340 | 0.6959 |
> | Collaboratively trained autoencoder | 0.8196 | 0.9368 | 0.8240 | 0.9342 |
>
> From the comparison with pretrained autoencoder, it is obvious that the autoencoder pretraining makes SMHGC to fall into local optimal very easily, due to the lack of information of single graph or single feature in these graph-dataset.
>
>
>
> > Additionally, the experimental settings should be described in more detail, including hyperparameters, learning rate, number of epochs, and so on.
>
> Thanks for your suggestion. We have included the detailed experimental settings in the appendix Table 5.
>
>
> | Parameter                        | ACM  | Texas | Chameleon | DBLP |
> | ---------------------------------| ---- | ----- | --------- | ---- |
> | $order$                          | 2    | 10    | 10        | 6    |
> | $\rho_h$                         | 0.5  | 3     | 3         | 3    |
> | $\rho_S$                         | 0.5  | 3     | 3         | 3    |
> | $k$                              | 300  | 30    | 250       | 300  |
> | Hidden dimension of $f_{x}$      | 128  | 128   | 1024      | 128  |
> | Output dimension of $f_{x}$      | 16   | 32    | 32        | 64   |
> | Hidden dimension of $f_{a}$      | 128  | 512   | 1024      | 256  |
> | Output dimension of $f_{a}$      | 16   | 128   | 32        | 128  |
> | Hidden dimension of $f_{\phi^v}$ | 128  | 128   | 1024      | 128  |
> | Output dimension of $f_{\phi^v}$ | 16   | 32    | 32        | 64   |
> | Number of layers $f_{x}$         | 2    | 4     | 4         | 2    |
> | Number of layers $f_{x}$ (repeated) | 2    | 4     | 4         | 4    |
> | Learning rate                    | 0.0005 | 0.001 | 0.005     | 0.005|
> | Weight\_decay                    | 0.0  | 0.0001| 0.005     | 5e-7 |
>
> *Table 5 summarizes the hyperparameters used for the datasets (ACM, Texas, Chameleon, and DBLP) in the experiments. The values are tailored for each dataset to achieve optimal performance, ensuring fair and consistent evaluation across different graph datasets.*
>
>
>
> ### 4. Impact of Discretization on Gradient Flow
>
> > To accommodate the discrete nature of the graph, the authors discretize the dense graph. However, this process may impact the gradient flow during training and thus hinder the model's performance. The authors should provide more details on how they handle this issue and how to achieve the statement "Note that the conducted aggregation in Eq. (10) can be replaced by any complicated GNNs" considering the differentiability of the model.
>
> Thanks for your detailed reviews and valuable suggestion. In our implementation, to ensure the backpropagation of gradients through the discretization operation used in  Eq. (8), we follow the generally used trick as implemented in *torch.nn.functional.gumbel_softmax*. Concretely, the code block of discretization operation in Eq. (8) are as below:
>
> ~~~python
> S_discret = torch.zeros_like(S_continual, memory_format=torch.legacy_contiguous_format)
> S_discret[top_k_indices] = 1.
> S_discret = S_discret - S_continual.detach() + S_continual
> ~~~
>
> This trick is generally used in many discretization operations and ensures the approximated gradients to be backpropagated successfully.
>
> We have included these details in our revisions. We hope this can address your concern.

---

> ### Author Response · Authors · 2024-12-06
> **Part4 Response**
>
> ### 5. Limited Experimental Comparisons
> > The experimental section includes comparisons with only six multi-view clustering methods across four datasets, which is relatively limited and insufficient to fully evaluate SMHGC's performance.
>
> Thanks for your suggestion. For data selection, we select four diverse widely used datasets following the community conventions, including:
>
> * Two widely used multi-view datasets ACM and DBLP.
> * Two widely used heterophilous datasets Texas and Chameleon.
>
> The real-world datasets could demonstrate the comparative results.
>
> In addition, we experiment on six synthesis MVHGC datasets following DuaLGR [r1]. The results can demonstrate the superiority performance of SMHGC targeting on MVHGC problem.
>
> In order to address your concern, we added extra three state-of-the-art MVC methods on ACM, DBLP for comparison as below:
>
> | ACM Dataset      | NMI  | ARI  | ACC  | F1   |
> | ---------------- | ---- | ---- | ---- | ---- |
> | R$^2$FGC         | 72.4 | 78.7 | 92.4 | 92.5 |
> | DIAGC            | 71.6 | 77.0 | 91.8 | 91.8 |
> | MFCGC            | 77.0 | 81.8 | 93.6 | 93.1 |
> | **SMHGC (ours)** | 81.1 | 83.2 | 93.9 | 93.9 |
>
>
> | DBLP Dataset     | NMI  | ARI  | ACC  | F1   |
> | ---------------- | ---- | ---- | ---- | ---- |
> | R$^2$FGC         | 50.8 | 56.3 | 81.0 | 80.5 |
> | DIAGC            | 78.3 | 83.7 | 93.3 | 92.8 |
> | MFCGC            | 79.0 | 84.3 | 93.5 | 93.1 |
> | **SMHGC (ours)** | 76.2 | 81.9 | 92.4 | 91.8 |
>
>
> | Texas Dataset    | NMI  | ARI  | ACC  | F1   |
> | ---------------- | ---- | ---- | ---- | ---- |
> | AHGFC            | 39.3 | 45.3 | 70.3 | 45.7 |
> | **SMHGC (ours)** | 41.8 | 46.9 | 71.3 | 49.8 |
>
> | Chemeleon Dataset | NMI  | ARI  | ACC  | F1   |
> | ----------------- | ---- | ---- | ---- | ---- |
> | AHGFC             | 21.5 | 16.2 | 43.2 | 41.6 |
> | **SMHGC (ours)**  | 20.0 | 15.1 | 42.1 | 41.3 |
>
>
>
> Thanks again for your suggestions. We hope the extended comparisons can address your concern.
>
> [r1] Yawen Ling, Jianpeng Chen, Yazhou Ren, Xiaorong Pu, Jie Xu, Xiaofeng Zhu, and Lifang He. Dual label-guided graph refinement for multi-view graph clustering. In Proceedings of the AAAI, pp. 8791–8798, 2023
>
> [r2] S. Yi, W. Ju, Y. Qin, X. Luo, L. Liu, Y. Zhou, and M. Zhang, “Redundancy-free self-supervised relational learning for graph clustering,” IEEE TNNLS, pp. 1–15, 2023.
>
> ### 6. Detailed Description of Synthetic Datasets
> > The authors should provide a more comprehensive description of the synthetic datasets used in the experiments.
>
> We appreciate your suggestions on the experiments.
>
> The synthetic datasets are provided in DuaLGR, the implementation details of these synthetic datasets are described as below, and we have included this in revisions.
>
> *Given the required HR, we divide all possible edges between nodes into two subsets based on the ground truth, i.e., homophilous and non-homophilous edges. Then the synthetic adjacent matrix is obtained by random sampling obeying a uniform distribution, whose number of edges is kept the same as the input view simultaneously.*

---

> ### Author Response · Authors · 2024-12-06
> **Part5 Response**
>
> > In Figure 4b, where results for some methods increase with rising HR values, this trend seems counter-intuitive, as higher heterophily typically complicates clustering, as the authors stated. The authors should clarify this phenomenon.
>
> Thank you for your careful reviews. As you mentioned, in Figure 4b, it is notable that the clustering performances of methods of O2MAC and MCGC start to increase when the heterophilous ratio achieves 0.7. This is actually a reasonable result and can be explained.
>
> *As Observation 2 illustrated, The HR of neighbor pattern similarity may be larger than the original graph, which implies that the larger heterophilous ratio of the original graph does not mean worse separability in Figure 4b. For example, when the heterophilous ratio achieves 1.0, the nodes with the same label connect to nodes in all other clusters, and these nodes with the same label would have the same neighbor pattern (i.e., connect to all the nodes in other clusters) when the edge number is large enough. Therefore, the clustering performance increases with the heterophilous ratio may be because of the increasing similarity of neighbor patterns.*
>
> We add this discussion in the revisions. In addition, to demonstrate this inference, we compute the HR of neighbor pattern similarity of the first view in synthetic dataesets. Specifically, we min_max normalize $AA^T$ at first, and set the threshold as the mean of the normalized neighbor pattern similarity, then, we regard the similarity value larger than this threshold as existing edges. HR is computed as the edge ratio of number of edges connecting nodes in the same cluster.
>
> | HR of original graph              | 0.5  | 0.6  | 0.7  | 0.8  | 0.9  | 1.0  |
> | --------------------------------- | ---- | ---- | ---- | ---- | ---- | ---- |
> | HR of neighbor pattern similarity | 0.38 | 0.34 | 0.33 | 0.34 | 0.38 | 0.44 |
>
> From the computed "HR of Neighbor pattern similarity", we can observe that the HR of neighbor pattern similarity starts to increase although HR of the original graph decreases when the HR of the original graph achieves 0.7. This observation substantiated our inference.
>
> ### 7. Analysis of Learned Similarity Weight Pairs
> > The authors should consider providing a detailed analysis of the learned similarity weight pairs across different datasets and HR levels.
> >
> > Since Table 2 indicates that the node feature similarity component has the greatest impact on SMHGC’s performance, visualizing the learned distributions could offer valuable insight into the respective contributions of node feature and neighbor pattern similarity.
>
> We greatly appreciate your valuable comments.  To address your concern, we have visualized the heatmap of learned *feature similarity* and *neighbor pattern similarity* in Figure 5 and Figure 6 for a homophilous graph in dataset ACM and a heterophilous graph dataset in Texas.
>
> The corresponding computed weight $\omega_x$ and $\omega_a$ for ACM and Texas are shown in the following table:
>
> | Dataset | $\omega_x$ | $\omega_a$ |
> | ------- | ---------- | ---------- |
> | ACM     | 1          | 0.12       |
> | Texas   | 1          | 0.26       |
>
> Comparing these values, we have the following conclusion:
>
> * **Compare the heatmaps between similarity in subspace and raw space** (i.e., $\mathbf{X}\mathbf{X}^\mathrm{T}$ and $\mathbf{Z}_x\mathbf{Z}_x^\mathrm{T}$,  $\mathbf{A}\mathbf{A}^\mathrm{T}$ and $\mathbf{Z}_a\mathbf{Z}_a^\mathrm{T}$): The feature similarity matrix $\mathbf{Z}_x\mathbf{Z}_x^\mathrm{T}$ significantly enhances homophilous information comparing to $\mathbf{X}\mathbf{X}^\mathrm{T}$ by learning more precise feature representations, resulting in a clear block structure, especially the comparison on heterophilous dataset Texas, where subspace heatmaps learns the cluster blocks from an undistinguishable heatmaps in raw space.
> * **Compare $\mathbf{Z}_x\mathbf{Z}_x^\mathrm{T}$and $\mathbf{Z}_a\mathbf{Z}_a^\mathrm{T}$,  $\omega_x$ and $\omega_a$**: On the both heterophilous and homophilous graph, the weight of feature similarity $\omega_x$ is greatly larger than the weight of neighbor pattern similarity $\omega_a$. This also explains the ablation study in Table 3 where node feature similarity component has the greatest impact. Focusing on the heatmaps, the blocks in $\mathbf{Z}_x\mathbf{Z}_x^\mathrm{T}$ is clearer than  $\mathbf{Z}_a\mathbf{Z}_a^\mathrm{T}$ for Texas, and slightly clearer for ACM.
> * **Comparing ACM and Texas:** 1) The weight of neighbor pattern similarity $\omega_a$ on Texas is larger than ACM, which may because that in heterophilous graph, the neighbor pattern similarity is more important than homophilous graph. 2) the heatmaps of ACM is clearer than heatmaps of Texas.

---

### Review · Reviewer_eMMK · 2024-11-18

**Summary Of Contributions:**

Since in the heterophilous graph, the connected nodes are no longer belong to the same class, the paper tries to build a new similarity matrix to replace the simple adjacency matrix in the message-passing procedure of GNN. This new similarity matrix is a linear combination of  $Z_xZ_x^T$ and $Z_aZ_x^T$, which are respectively the approximation of $AA^T$ and $XX^T$ where $A$ is the adjacency matrix and $X$ is the node features. The experiments are conducted on multi-view heterophilous graph clustering datasets and obtain good empirical performances.

**Audience:**

No

**Claims And Evidence:**

No

**Requested Changes:**

- Clarify the rationale for using approximations of $ Z_xZ_x^T$ and $ Z_aZ_a^T$ rather than the original similarity matrices. Justify why these approximations are preferable or necessary.
- Elaborate on the purpose and methodology of using $\overline{H}$ for computing weights or scores.
- Correct the notations mentioned.
- Reassess the definition of *Neighbor pattern similarity* $AA^T$ in light of existing graph theory concepts. If the definition diverges from standard interpretations (e.g., adjacency matrix powers for walks), provide explicit reasoning and context.
- Clarify the unclear notions mentioned in weaknesses, mathematically if possible.
- Reconsider the inclusion of the generalization analysis in Section 3.6. If it is not directly relevant to the main objectives of the paper, move it to supplementary materials or explain its relevance to the core contribution.
- Report the variance of results for other methods on the ACM dataset to ensure a fair comparison with the proposed method. Discuss the potential causes for the large variance in the results for ACM.
- Address the critique regarding the contribution's focus on the single-view level. Clarify how the proposed method advances the state-of-the-art in multi-view learning or explicitly position the work as a single-view contribution, adjusting the title and claims accordingly.
- Conduct a thorough proofreading of the manuscript to address general readability issues.

**Strengths And Weaknesses:**

Strengths:
* Good empirical results on real-world graph datasets.
* The idea of having a new similarity matrix (instead of using the adjacency matrix) for the heterophilous graphs to help the node clustering is logical.

Weaknesses:
* The motivations for some choices of model design are confusing and not convincing.
  * If the paper supposes $Z_xZ_x^T$ and $Z_aZ_a^T$ are good similarity matrices representing homophilous information, why use their approximations instead of themselves?
 * The institution of using the final embedding $\overline{H}$ to compute weights or scores is not clear and lacks the development.
* The paper is generally not well-written. Here are some examples:
  * In Definition 2, $\tilde{E}$ should be $\mathbb{R}^{N\times d} \times  \mathbb{R}^{N\times N} \to \mathbb{R}^{N\times N}$ instead of $\mathbb{R}^{N\times d} ,\mathbb{R}^{N\times N} \to \mathbb{R}^{N\times N}$
  * The dimensions of $Z_a$, and $Z_x$ are not indicated, which is not clear to me.
  * When a function have several variables, please use $f(\cdot, \cdot)$ instead of $f(\cdot; \cdot)$
  * The norm operator on page 8 is not defined.
  * Equation 12 is unclear to us about how parameters to learn are involved in the objective function.
* In Definition 4, the authors define a new notion called *Neighbor pattern similarity* by $AA^T$, however, it's known to us that the power of the adjacency matrix $A^n$ gives the number of the number of walks of length $n$ between nodes.
* Many notions introduced in the paper are not clearly or mathematically presented:
  * How you define mathematically *an idea subspace* in page 6?
  * On page 7, what do you mean by HR of neighbour patterns? and how do you increase it?
  * In section 3.5, what do you mean by the soft and target distribution of $\overline{H}$?
  * In section 4.3 ablation study, what do you mean by *absence of $\mathcal L_{sim}$* and the *absence of $\mathcal L_{r}$*? Do you mean that $z_a$, $z_x$ or $z_f$ are randomly initialized and still used to compute the final embedding?
* The generalization analysis in section 3.6 is not relevant to the main objective of the paper. I don't see the relevance of putting it in the main paper.
* The variance of results on ACM is very big, however, the variance of other methods is not reported on ACM.
* The main contribution of the paper is on the single-view level instead of the multi-view level.

---

> ### Author Response · Authors · 2024-12-06
> **Part1 Response**
>
> We appreciate your reviews and constructive suggestions. The corresponding revisions conducted in the manuscript are colored in $\color{darkred}{darkred}$.
>
> ### 1. Rationale for Approximations of $Z_x{Z_x}^T$ and $Z_a{Z_a}^T$
>
> > Clarify the rationale for using approximations of  $Z_x{Z_x}^T$ and $Z_a{Z_a}^T$ rather than the original similarity matrices. Justify why these approximations are preferable or necessary.
>
> We appreciate your comments and suggestions. To address your concern, the heatmap visualization of  $\mathbf{Z}_x \mathbf{Z}_x^T$ and $\mathbf{Z}_a\mathbf{Z}_a^T$ are included in Figures 5 and 6 of revised manuscripts, which clearly demonstrates the effectiveness of approximation of  $\mathbf{Z}_x \mathbf{Z}_x^T$ and $\mathbf{Z}_a\mathbf{Z}_a^T$.
>
> In addition, we intuitively analyzed the rationale for approximations of $Z_x{Z_x}^T$ and $Z_a{Z_a}^T$ as below.
>
> **Intuition 1 [Multi-view collaborative optimization]:**  The information in $XX^T$ and $AA^T$  is view specific. A key idea for multi-view clustering is to collaboratively optimize the representations of each view. Therefore, approximation of $z_x{z_x}^T$ and $z_a{z_a}^T$ provide a chance to include the information from other views during the optimization process by the backpropagation of final KL loss ($\mathcal{L}_{KL}$) [r1, r2].
>
> **Intuition 2 [Learn a better subspace representation]:** A traditional autoencoder is only optimized by $\mathcal{L}_r$, which has been proved to be effective in multi-view subspace learning [r2, r3]. In order to learn the similarity information in subspace, a natural way is to introduce the similarity regularization terms ($\mathcal{L}{sim}$) for subspace learning.
>
> On the other hand,  From observation 2 (Figure 1. (b)), it can be seen that the neighbor pattern similarity matrix or feature similarity matrix is not always the most homophilous graphs, e.g., in ACM dataset. Therefore, direct use of $XX^T$ and $AA^T$ may introduce inevitable non-homophilous information. So we project the original  $XX^T$ and $AA^T$ to a subspace by an autoencoder-like structure, which can not only preserve most important similarity information (by regularization terms $\mathcal{L}_{sim}$)and feature information (by $\mathcal{L}_r$), but also obtain the guidance from other views [r3, r4]. Figures 5 and 6 also demonstrate the clearer distinguishability of approximation of $\mathbf{Z}_x \mathbf{Z}_x^T$ and $\mathbf{Z}_a\mathbf{Z}_a^T$.
>
> In addition, as hypothesis 1 stated, *Hypothesis 1: In node clustering tasks, the higher the similarity between two nodes in an ideal subspace, the greater the likelihood that they belong to the same cluster*, we expect the approximations of $z_x{z_x}^T$ and $z_a{z_a}^T$ to learn a better subspace for better similarity approximation with this design, guiding better clustering results.
>
>
>
> Following your suggestions, we added the following discussion on the rationale for approximation of  $\mathbf{Z}_x \mathbf{Z}_x^T$ and $\mathbf{Z}_a\mathbf{Z}_a^T$ to Section 3.2 shown as $\color{darkred}{DarkRed}$, to clarify the motivation for autoencoder like structure.
>
> *The autoencoder design for projecting $\mathbf{A}$ to $\mathbf{Z_a}$ by $f_a(\cdot)$ and $\mathbf{X}$ to $\mathbf{Z_x}$ by $f_x(\cdot)$ has two main goals: (1) Multi-view collaborative optimization: the backpropagation of final multi-view clustering loss ($\mathcal{L}_{KL}$) introduces clustering information from other views [r1, r2], achieving multi-view collaborative optimization for each single view's similarity matrices. (2) Learn a better subspace: as stated in Hypothesis 1, the similarity matrices are expected to be constructed in an ideal subspace. The projection of $\mathbf{Z}_x \mathbf{Z}_x^T$ and $\mathbf{Z}_a\mathbf{Z}_a^T$  can not only preserve most important similarity information (implemented by regularization term $\mathcal{L}{sim}$) and feature information (by $\mathcal{L}_r$), but also obtain the guidance from other views [r3, r4], resulting in the possibility of learning a ideal subspace.*
>
> [r1] P. Zhu, X. Yao, Y. Wang, B. Hui, D. Du and Q. Hu. Multiview Deep Subspace Clustering Networks. in *IEEE Transactions on Cybernetics*, vol. 54, no. 7, pp. 4280-4293, July 2024.
>
> [r2] Xu J, Ren Y, Li G, et al. Deep embedded multi-view clustering with collaborative training[J]. Information Sciences, 2021, 573: 279-290.
>
> [r3] Peng X, Feng J, Xiao S, et al. Structured autoencoders for subspace clustering[J]. IEEE Transactions on Image Processing, 2018, 27(10): 5076-5086.
>
> [r4] Baldi P. Autoencoders, unsupervised learning, and deep architectures[C]//Proceedings of ICML workshop on unsupervised and transfer learning. JMLR Workshop and Conference Proceedings, 2012: 37-49.

---

> ### Author Response · Authors · 2024-12-06
> **Part2 Response**
>
> ### 2. Purpose and Methodology of Using $\mathbf{\overline{H}}$
>
> > Elaborate on the purpose and methodology of using $\mathbf{\overline{H}}$ for computing weights or scores.
>
> Thank you for your valuable feedback. To address your concerns regarding the purpose and methodology of using the final embedding $\mathbf{\overline{H}}$ for computing weights or scores, we have emphasized our explanations about the objectives of weighting embeddings from each view ($\omega^v_h$ for $\mathbf{H}^v$) and similarity matrices within each view ($\omega^v_{a,x}$ for $\mathbf{A}_a$ and $\mathbf{A}_x$). Below, we provide a elaboration on both the purpose and methodology. In addition, we have highlighted these details in **Section 3.3** and **Section 3.4** of our manuscript.
>
> **Purpose of Computing Weights and Scores**
>
> 1. **For Embeddings from Each View ($\omega^v_h$ for $\mathbf{H}^v$):**
>    - Different views contribute differently to the final clustering outcomes. Without appropriate weighting, poor-quality views can disproportionately affect the results. By assigning weights based on the quality of each view, we can mitigate the impact of less informative views and enhance the influence of more relevant ones.
>
> 2. **For Similarity Matrices in Each View ($\omega^v_{a,x}$ for $\mathbf{A}_a$ and $\mathbf{A}_x$):**
>    - Feature similarity and neighbor pattern similarity carry distinct homophilous information. Assigning higher weights to matrices with richer homophilous content ensures that such information is prioritized during the clustering process.
>
> **Why Use $\mathbf{\overline{H}}$ for Weighting?**
>
> - **Challenge:** In clustering tasks, true labels are not available, making it difficult to evaluate the quality of embeddings or similarity matrices directly.
> - **Solution:** Multi-view consensus information in final embedding $\mathbf{\overline{H}}$ serves as a proxy for labels. This pseudo-label approach guides the learning of high-quality representations for individual views, which, when combined, yield a superior multi-view representation. The iterative refinement process enhances clustering performance.
>
> **Methodology of Using $\overline{\mathbf{H}}$**
>
> 1. **Weighting Embeddings from Each View ($\omega^v_h$ for $\mathbf{H}^v$):**
>
>    - We calculate the cosine similarity $score^v$ between  $\mathbf{H}^v$ and $\mathbf{\overline{H}}$, then normalize this score using Equation (11):
>      $$
>      \omega_h^v = \left(\frac{score^v}{\max{(score^1, score^2, \cdots, score^V)}}\right)^\rho,
>      $$
>
> - The underlying principle is that embeddings closer to the multi-view consensus embedding are considered of higher quality and thus receive greater weight.
>
> 2. **Weighting Similarity Matrices in Each View ($\omega^v_{a,x}$ for $\mathbf{A}_a$ and $\mathbf{A}_x$):**
>
>    - We determine the cosine similarity between the similarity matrix $\mathbf{A}_a$ and $\mathbf{A}_x$ and $\omega^v_a$, $\omega^v_x$, the multi-view consensus similarity matrix $\overline{\mathbf{H}}\overline{\mathbf{H}}^\mathrm{T}$, followed by normalization according to Equation (6):
>      $$
>      (\omega_x, \omega_a) = \text{norm}\left(sim(\mathbf{A}_x; \overline{\mathbf{H}}\overline{\mathbf{H}}^\mathrm{T}), sim(\mathbf{A}_a; \overline{\mathbf{H}}\overline{\mathbf{H}}^\mathrm{T})\right),
>      $$
>
>    - The rationale here is that similarity matrices more aligned with the multi-view consensus contain more homophilous information and therefore deserve higher weights.
>
> ### 3. Correction of Notations
>
> Thanks for your careful reviews. We have corrected all notations as you mentioned.

---

> ### Author Response · Authors · 2024-12-06
> **Part3 Response**
>
> ### 5. Clarification of Unclear Notions
>
> > Clarify the unclear notions mentioned in weaknesses, mathematically if possible.
>
> We appreciate your careful reviews. We have clarified the notations as you suggested in the revised manuscript, and we also put them below for your reference.
>
>
>
> > * How you define mathematically *an idea subspace* in page 6?
>
> Thank you for your valuable feedback. We appreciate your suggestion to clarify the definition of the ideal subspace in the context of Hypothesis 1. According to your suggestion, we have mathematically defined the ideal subspace in page 22, Appendix. D.2 Mathematical Definition and Discussion about Hypothesis 1.
>
> More over, we have also added this detailed explanation of ideal subspace to the revised manuscript and included the visualization of subspace in Figures 5 and 6, which can demonstrate the effectiveness of the usage of ideal subspace.
>
>
> > * On page 7, what do you mean by HR of neighbor patterns? and how do you increase it?
>
> Thanks for your clarification. The HR of neighbor patterns refers to the HR of neighbor pattern similarity following the definition of original HR [r8].
>
> We have revised the HR of neighbor patterns to HR of neighbor pattern similarity, and included the implementation details on how we increasing the HR of neighbor pattern similarity.
>
> **Formal Definition of HR of neighbor pattern similarity:**
>
> Following previous works, HR (Homophilous Ratio) is defined as the proportion of edges that connect nodes in a same cluster:
> $$
> \text{HR} = \frac{1}{|\mathcal{E}|} \sum_{(i, j) \in \mathcal{E}} \delta(\mathbf{x}_i, \mathbf{x}_j),
> $$
> where $\mathcal{E}$ is the set of edges in the graph, $\mathbf{x}_i$ and $\mathbf{x}_j$ are the feature vectors of nodes $i$ and $j$, and $\delta(\mathbf{x}_i, \mathbf{x}_j)$ is an indicator function that equals 1 if $\mathbf{x}_i$ and $\mathbf{x}_j$ are similar (e.g., based on a predefined similarity threshold) and 0 otherwise.
>
> In the context of neighbor pattern similarity, the computation of HR conducts on neighbor pattern similarity matrix $\mathbf{A}\mathbf{A}^\mathrm{T}$, $\mathbf{x}_i$ and $\mathbf{x}_j$ refer the set of neighbors of node $i$ and node $j$. Specifically,
>
> - Let $\mathbf{N}_i$ denote the set of neighbors of node $i$, and let $c_i$ be the class label of node $i$. The neighbor pattern similarity between nodes $i$ and $j$ can be measured by the similarity of their neighbor sets $\mathbf{N}_i$ and $\mathbf{N}_j$.
>
> - The HR of neighbor pattern similarity can be defined as:
>   $$
>   \text{HR} = \frac{1}{|\mathcal{E}|} \sum_{(i, j) \in \mathcal{E}} \delta(\mathbf{N}_i, \mathbf{N}_j),
>   $$
>   where $\delta(\mathbf{N}_i, \mathbf{N}_j)$ is an indicator function that equals 1 if the neighbor sets $\mathbf{N}_i$ and $\mathbf{N}_j$ are similar (e.g., based on a predefined similarity threshold 0.5) and 0 otherwise.
>
>
> **How to increase HR of neighbor pattern similarity:**
>
> To increase the HR of neighbor pattern similarity, we leverage the insight from Proposition 1, which suggests that "good heterophily" corresponds to similar neighbor pattern distributions. Our approach involves modeling the neighbor distribution of each class as a Gaussian distribution and reducing the standard deviation (std) of these distributions. This reduction in std enhances the HR of neighbor pattern similarity. Here is a two steps description of the method:
>
> 1. **Modeling Neighbor Distributions:**
>    - Assume that the neighbor distribution of nodes within each class follows a Gaussian distribution. Let $\mathcal{N}(\mu_c, \sigma_c^2)$ represent the Gaussian distribution for class $c$, where $\mu_c$ is the mean and $\sigma_c$ is the std.
> 2. **Reducing Standard Deviation:**
>    - By reducing the standard deviation $\sigma_c$, we make the neighbor distributions more concentrated around their means. This concentration increases the likelihood that nodes within the same class will have similar neighbor patterns, thereby enhancing the homophilous ratio (HR).
>
> To explain how the std influences HR, we give the following examples:
>
> - **High Standard Deviation ($\sigma_c$ is large):** When the std is large, the neighbor distribution becomes more spread out, leading to a more uniform distribution. In this case, nodes are likely to connect to other nodes randomly, resulting in a lower HR.
> - **Low Standard Deviation ($\sigma_c$ is small):** When the std is small, the neighbor distribution becomes more concentrated. Nodes within the same class are more likely to connect to a specific subset of nodes, leading to higher similarity in neighbor patterns and a higher HR.
>
> By reducing the std of the neighbor distributions, we can effectively increase the HR of neighbor pattern similarity as shown in Figure 3.

---

> ### Author Response · Authors · 2024-12-06
> **Part4 Response**
>
> > * In section 3.5, what do you mean by the soft and target distribution of $\mathbf{\overline{H}}$?
>
> Thanks for your notations on this description. The illustration about $\mathbf{\overline{Q}}$ and $\mathbf{\overline{P}}$ are not accurate. We have revised the description of "soft and target distribution" to “soft and target cluster assignments”. The concrete definition for "soft and target cluster assignments" is proposed in [r9]. More details are included in the revisions Appendix. D.1 Kullback-Leibler Divergence Loss.
>
> > * In section 4.3 ablation study, what do you mean by *absence of Lsim* and the *absence of Lr*? Do you mean that za, zx or zf are randomly initialized and still used to compute the final embedding?
>
> In the context of our ablation study presented in Section 4.3, the terms "w/o $\mathcal{L}\text{sim}$" and "w/o $\mathcal{L}_{{r}}$" refer to the exclusion of specific loss components from the overall optimization process.
>
> **Impact and meaning of "w/o $\mathcal{L}\text{sim}$" and "w/o $\mathcal{L}_{{r}}$"**
> * When we say w/o $ \mathcal{L}{\text{sim}}$, it means that during the optimization process, we only use the remaining loss terms, specifically, $\mathcal{L}r $ and $\mathcal{L}\text{KL}$, to update the embeddings $z_a$ and $z_x$.  Without $\mathcal{L}{\text{sim}}$, the embeddings $z_a$ and $z_x$ will not be directly influenced by the original similarity information. Instead, they will be optimized based on the reconstruction loss $\mathcal{L}{\text{r}}$ and the KL divergence loss $\mathcal{L}{\text{KL}}$. This can lead to embeddings that may not preserve the original similarity relationships as effectively.
> * When we say w/o $\mathcal{L}r$, it means that during the optimization process, we only use  $\mathcal{L}{\text{sim}}$ and  $\mathcal{L}{\text{KL}}$ to update the embeddings  $z_a$ and $z_x$. Without $\mathcal{L}r$, the embeddings $z_a$ and $z_x$ will not be optimized for reconstructing the input data. They will primarily focus on preserving the similarity relationships (if $\mathcal{L}{\text{sim}}$ is present) and maintaining the distribution consistency (via $\mathcal{L}{\text{KL}}$). This can result in embeddings that are less effective in reconstructing the original data but may still capture some similarity information.
>
>  **Initialization and Optimization of $z_a$, $z_x$, and $z_f$**
>
> - **Initialization**: The embeddings $z_a$, $z_x$, and $z_f$ are randomly initialized.
> - **Optimization**:
>   - **With   $L_{\text{sim}}$ and  $L_{\text{KL}}$**: If $L_{\text{sim}}$ is present, $z_a$, and $z_x$ are optimized to preserve the original similarity information while also maintaining distribution consistency.
>   - **With $L_{\text{r}}$ and $L_{\text{KL}}$**: If  $L_{\text{r}}$ is present,  a $z_a$, and $z_x$ are optimized to reconstruct the input data accurately while also maintaining distribution consistency.
>
> [r8] Xin Zheng, etc. Graph neural networks for graphs
> with heterophily: A survey. arXiv preprint arXiv:2202.07082, 2022.
>
> [r9] J. Xie, etc. Unsupervised deep embedding for clustering analysis. in ICML, 2016, pp. 478–487.
>
>
>
> ### 6. Inclusion of Generalization Analysis
>
> > Reconsider the inclusion of the generalization analysis in Section 3.6. If it is not directly relevant to the main objectives of the paper, move it to supplementary materials or explain its relevance to the core contribution.
>
> Thanks for your valuable comments. According to your suggestion, we have moved this Section to the Appendix.
>
> ### 7. Variance Reporting for ACM Dataset
>
> > Report the variance of results for other methods on the ACM dataset to ensure a fair comparison with the proposed method. Discuss the potential causes for the large variance in the results for ACM.
>
> Thanks for your suggestions. The results about ACM and DBLP reported in Table 2 are directly drawn from their original papers, where the variance is omitted. For the proposed method SMHGC, we run it five times on each dataset and compute their variance. According to your suggestion, we have re-implemented the two most recent SOTAs, DuaLGR and BTGF to show their variance for comparison.
>
> **ACM：**
>
> | Method       | NMI (%)        | ARI (%)        | ACC (%)        | F1 (%)         |
> | ------------ | -------------- | -------------- | -------------- | -------------- |
> | DuaLGR       | 73.1 ± 1.1     | 78.9 ± 0.7     | 92.5 ± 0.2     | 92.5 ± 0.2     |
> | BTGF         | 68.7 ± 0.0     | 73.6 ± 0.0     | 90.4 ± 0.0     | 90.5 ± 0.0     |
> | SMHGC (ours) | **81.1 ± 4.1** | **83.2 ± 5.2** | **93.9 ± 2.0** | **93.9 ± 2.0** |
>
> **DBLP：**
>
> | Method       | NMI (%)        | ARI (%)        | ACC (%)        | F1 (%)         |
> | ------------ | -------------- | -------------- | -------------- | -------------- |
> | DuaLGR       | 75.4 ± 0.0     | 81.6 ± 0.0     | 92.3 ± 0.0     | **91.8 ± 0.0** |
> | BTGF         | 45.8 ± 0.0     | 49.0 ± 0.0     | 74.6 ± 0.0     | 73.6 ± 0.0     |
> | SMHGC (ours) | **76.2 ± 0.8** | **81.9 ± 0.2** | **92.4 ± 0.2** | **91.8 ± 0.2** |

---

> ### Author Response · Authors · 2024-12-06
> **Part5 Response**
>
> ### 8. Focus on Single-View Contribution
>
> > Address the critique regarding the contribution's focus on the single-view level. Clarify how the proposed method advances the state-of-the-art in multi-view learning or explicitly position the work as a single-view contribution, adjusting the title and claims accordingly.
>
> Thank you for your insightful comments. We appreciate your feedback and would like to clarify the role of single-view and multi-view aspects in our work.
>
> 1. **Motivation and Observations from Single-View:**
>    - It is true that the initial motivation and observations are derived from single-view scenarios. This foundational understanding helps us identify key challenges and opportunities in data representation and clustering.
>
> 2. **Main Contributions Based on Multi-View Consensus:**
>    - Despite the single-view origins, the main contributions of our work are firmly rooted in multi-view learning. The proposed similarity terms for subspace learning and the overall optimization framework are designed to leverage multi-view consensus.
>    - Specifically, the combination of the proposed neighbor pattern similarity and feature similarity matrix is crucial for capturing comprehensive relationships across multiple views. This integration ensures that the model benefits from the rich, complementary information available in multi-view data.
>
> 3. **Multi-View Clustering Task Specificity:**
>    - Each component of our framework is tailored for multi-view clustering tasks. For instance, the neighbor pattern similarity term is designed to capture consistent neighborhood structures across different views, which is essential for robust clustering.
>    - Similarly, the feature similarity matrix is constructed to reflect the inherent relationships between features across multiple views, enhancing the model's ability to identify meaningful clusters.
>
> 4. **Label Agnostic Context:**
>    - In the context of label-agnostic learning, the approximation of the homophilous ratio for each matrix becomes particularly challenging in a single-view setting. Our approach effectively addresses this by leveraging multi-view consensus information.
>    - By combining information from multiple views, we can more accurately estimate the homophilous ratio, leading to improved clustering performance even without labeled data.
>
> 5. **Clarification of Contribution Scope:**
>    - To better align with the reviewer's feedback, we have added the discussion on how our methods contribute to MVHGC problems.
>    - Additionally, we will provide a detailed discussion in the introduction and related work sections to highlight the unique advantages of our multi-view approach over single-view methods.
>
> We hope these clarifications address your concerns and provide a clearer understanding of our work.

---

### Review · Reviewer_JU9m · 2024-11-19

**Summary Of Contributions:**

The paper tackles the problem of unsupervised node classification where label information is unavailable during training. The primary contribution is a proposed method, SMHGC, which integrates feature similarity, neighbor pattern similarity, and multi-view global similarity to generate node embeddings. Experiments conducted on four datasets against ten baselines showcase the proposed method's superiority, particularly on heterophilous graphs.

**Audience:**

Yes

**Claims And Evidence:**

Yes

**Requested Changes:**

See Strengths and weakness.

**Strengths And Weaknesses:**

---

### Strengths

1. **Thorough Analysis**:  The method is extensively evaluated, with clear visualizations of experimental results and an ablation study demonstrating the contributions of each module and loss term.

2. **Illustrative Plots**:  The figures in the introduction help motivate the method and provide a visual understanding of the challenges and proposed solutions.

3. **Clarity in Writing**:  The paper is well-organized and easy to read, making the technical details accessible to readers.

---

### Weaknesses and Suggestions for Improvement

#### **Introduction**
1. **Task Definition**:
   Clearly define the task as node-level classification in both the abstract and the early part of the introduction. This will help readers understand the study's scope immediately.
   - In Section 2.1, use formal notations to define the task and specify whether the graph views are provided or created using preprocessing.

2. **Clarification for Figures**:
   - **Figure 1(a)**: Add an explanation of how graphs with varying heterophily ratios are generated.
   - **Figure 1(b)**: Provide a brief description of the labels (e.g., "acm_a1", "acm_a2") on the x-axis. It seems these are derived from ACM and DBLP datasets, but this should be clarified.

---

#### **Preliminaries**
1. **Homophily Definition**:
   The definition of homophily (Definition 1) is similar to the concept of edge homophily from [1], and this connection should be explicitly acknowledged or cited as the definition of homophily and heterophily is not the contribution of the current work. Furthermore, a more complexed scenario of homophily and heterophily is proposed in [2], add disccusion to clarify the scope of the current research.

---

#### **Proposed Method**
1. **"Good Heterophily"**:
   The concept defined in Proposition 1 overlaps with "cross-class neighborhood similarity (CCNS)" introduced in [3]. Discuss the differences or provide appropriate citations to avoid confusion.

2. **Encouraging Original Representations**:
   - The loss terms \( L_{\text{sim}_a} \) and \( L_{\text{sim}_x} \) aim to align \( Z_aZ_a^T \) with \( AA^T \) and \( Z_xZ_x^T \) with \( XX^T \). Would this lead the model to just use the original adjacency matrix \( A \) and feature matrix \( X \)?
   - Include an ablation study comparing the learned embeddings \( Z_a \) and \( Z_x \) with \( A \) and \( X \). If these are significantly different, explain why this occurs.

3. **Metrics and Design Choices**:
   - Describe the metrics used in this paper NMI and HR, including whether higher or lower values are better.
   - Explain the "parameter-free message passing layer" referenced in Figure 3.

4. **Clustering and Homophily**:
   - Elaborate on why the increase in homophilous information fails to improve clustering performance via a message-passing layer (e.g., the blue line in Figure 3).

5. **Computation of \( P \) and \( Q \)**:
   Provide a detailed explanation of how \( P \) and \( Q \) are computed in Section 3.5.

---

#### **Experiments**
1. **Baseline Descriptions**:
   The baselines used for comparison are not described. explain why are they included in this work and provide a brief summary of their methodologies.

2. **Explanation of Table 1**:
   Define the terms like "hr 0.82 & 0.64" and other abbreviations used in Table 1.

3. **Performance Discussion**:
   - Section 4.2 lacks detailed analyse into why SMHGC outperforms other baselines on heterophilous graphs. Provide a deeper discussion of the model's strengths compared to existing methods.
   - state concretly the limitations of traditional approaches which would explain why they fail. And explain why SMHGC excels in scenarios with heterophily

---

### References
Ensure the following references are appropriately cited and discussed:

1. **[1] Beyond Homophily in Graph Neural Networks: Current Limitations and Effective Designs**
   Zhu et al., 2020.

2. **[2] Multi-label Node Classification On Graph-Structured Data**
   Zhao et al., 2023.

3. **[3] Is Homophily a Necessity for Graph Neural Networks?**
   Ma et al., 2021.

---

> ### Author Response · Authors · 2024-12-06
> **Part1 Response**
>
> Thank you for your reviews. We respond to your comments question-by-question below, and the corresponding revisions conducted in the manuscript are colored in $\color{blue}{blue}$.
>
> **#1** Clearly define the task as node-level classification in both the abstract and the early part of the introduction. This will help readers understand the study's scope immediately.
>
> **[response]**:The task in this paper is a node-level clustering task, and the specific problem definition is provided in Problem 1 in Section 2.1. Your suggestion is very helpful, and we have updated basic definition in the abstract and introduction. The modification is as follows:
>
> > Abstract:
> >
> > With the increasing prevalence of graph-structured data, multi-view graph clustering, which aims to partition nodes into distinct groups by leveraging complementary information from multiple graph views representing different aspects of the same sample, has become a fundamental technique in various applications.
> >
> > Introduction:
> >
> > Deep multi-view graph clustering (MVGC), which aims to partition nodes into distinct groups by integrating complementary and consistent information from multiple graph views representing different aspects of the same data, has recently been explored to address this problem under unsupervised settings.
>
>
>
>
>
> **#2** In Section 2.1, use formal notations to define the task and specify whether the graph views are provided or created using preprocessing.
>
> **[response]**: The task definition is in Problem 1 in Section 2.1.  The graph is given by the dataset rather created. Please find the details of datasets in Appendix Table 4. To ease understanding, we have modified it as follows:
>
> > Specifically, given $V$ different views with graphs $\{\mathcal{G}^v\}_{v=1}^V$ and features $\{\mathbf{X}^v\}_{v=1}^V$, which are assumed to be provided as input, the goal of MVGC is to partition the nodes into different classes. We formally define this task in Problem 1.
>
>
>
> **#3** Clarification for Figures.
>
> - Figure 1(a): Add an explanation of how graphs with varying heterophily ratios are generated.
>
>   **[response]**: Graphs with varying heterophily ratios were generated by adjusting edge connections in a ACM base graph, progressively increasing the proportion of edges between nodes of different classes to control the heterophily ratio. According to you suggestion, the explanation has been added in the caption of Figure 1 as follows:
>
>   > Graphs with varying heterophily ratios were generated by adjusting edge connections in a ACM base graph.
>
>   In addition, we included a detailed description about the data and model selection in Figure 1(a):
>
>   > **Experimental details of Observation 1**
>   >
>   > In Figure 1(a), we explored the inherent Limitation of GNNs for Clustering on Heterophilous Graph. The experiments follows the following settings, expecting to lead a robust Observation.
>   >
>   > * **Model selection**: to exclude the influential factors from model design, and optimization influence, we design a very basic GNN that contains only two non-parametric message passing layers. The basic GNN can be formulated as below. After $\mathbf{h}^2$ is obtained, we conduct K-means to get final clustering results and compute its NMI scores.
>   >   $$
>   >   \mathbf{h}^1_i = \mathbf{x}_i + \sum_j\mathbf{x}_j,$$ $$
>   >   \mathbf{h}^2_i = \mathbf{h}^1_i + \sum_j{\mathbf{h}^1_j}, j \in \mathcal{N}(i)
>   >   $$
>   >
>   > * **Data selection**: we choose two most widely used graph dataset Cora and Citeseer, and refine the two graphs to obtain different homophilous ratio. Specifically, to control their homophilous ratio, we keep their edge number to be unchanged, and reorganize their edges by randomly dropping homophilous edges and adding new homophilous edges.
>
> - Figure 1(b): Provide a brief description of the labels (e.g., "acm_a1", "acm_a2") on the x-axis. It seems these are derived from ACM and DBLP datasets, but this should be clarified.
>
>   **[response]**: The detailed description of the datasets is presented in Table 3."ACM_A1" and "ACM_A2" correspond to the first and second views of the dataset, respectively. We have added a detailed explanation in the caption of Figure 1, and the modification is as follows:
>
>   > Each entry in the dataset corresponds to a view in Table 3. For example, "ACM_A1" corresponds to $\mathcal{G}^1$ in Table 4.

---

> > ### Comment · Reviewer_JU9m · 2024-12-19
> > **Thank the authors**
> >
> > I thank the authors for incorporating the necessary descriptions, definitions, and clarifications highlighted in the initial review. However, the motivation and rationale behind using the approximations $Z_aZ_a^T$ and $Z_xZ_x^T$, guided by $AA^T$ and $XX^T$, remain unclear. I encourage the authors to investigate and explain why employing $AA^T$ and $XX^T$ as guiding signals results in representations of even higher quality than the guiding signals themselves.

---

> > > ### Author Response · Authors · 2024-12-27
> > > **Response for $AA^T$ and $XX^T$ as Guiding Signals**
> > >
> > > We greatly appreciate you for your response on our revisions, and delightedly to see this revisions can resolve most your concerns. We answer your new questions as follows in detail.
> > >
> > > **Q:** Why employing $AA^T$ and $XX^T$ as guiding signals results in representations of even higher quality than the guiding signals themselves.
> > >
> > > **[response]**: Thanks for your constructive suggestions. Reviewer #eMMK raised the similar questions. Therefore, in the revisions, we thoroughly analyzed the explains why we employing $AA^T$ and $XX^T$ as guiding signal rather than directly use themselves. Conducting $AA^T$ and $XX^T$ as guiding signal implements subspace learning where the rationale is supported in Hypothesis 1, i.e.,  we need to approximate an ideal subspace. The details are as follows:
> > >
> > >
> > >
> > > **1. The fact that subspace representation $ZZ^T$ performs better than guiding signals.** In the revisions, we experimentally demonstrated this in Section 4.5. The similarity matrices in  Figures 5 and 6 show that heatmaps of learned subspace representations ${Z_aZ_a} ^T$  and ${Z_xZ_x} ^T$ exhibit clearer cluster boundaries than guiding signals $AA^T$ and $XX^T$.
> > >
> > >
> > >
> > > **2. Intuitive explanations.** We also intuitively explains why the subspace representation can obtain better performance than guiding signals themselves as follows:
> > >
> > > **Intuition 1 [Multi-view collaborative optimization]:**  The information in $XX^T$ and $AA^T$  is view specific. A key idea for multi-view clustering is to collaboratively optimize the representations of each view. Therefore, approximation of $z_x{z_x}^T$ and $z_a{z_a}^T$ provide a chance to include the information from other views during the optimization process by the backpropagation of final KL loss ($\mathcal{L}_{KL}$) [r1, r2].
> > >
> > > **Intuition 2 [Learn a better subspace representation]:** A traditional autoencoder is only optimized by $\mathcal{L} _ r$, which has been proved to be effective in multi-view subspace learning [r2, r3]. In order to learn the similarity information in subspace, a natural way is to introduce the similarity regularization terms ($\mathcal{L}_{sim}$) for subspace learning.
> > >
> > > On the other hand,  From observation 2 (Figure 1. (b)), it can be seen that  the neighbor pattern similarity matrix or feature similarity matrix is not always the most homophilous graphs, e.g., in ACM dataset. Therefore, directly use of $XX^T$ and $AA^T$ may introduce inevitable non-homophilous information. So we project the original  $XX^T$ and $AA^T$ to a subspace by an autoencoder-like structure, which can not only preserve most important similarity information (by regularization terms $\mathcal{L}_{sim}$) and feature information (by $\mathcal{L}_r$), but also obtain the guidance from other views [r3, r4]. Figures 5 and 6 also demonstrate the clearer distinguishability of approximation of $Z_x{Z_x}^T$ and $Z_a{Z_a}^T$.
> > >
> > > In addition, as hypothesis 1 stated, *Hypothesis 1: In node clustering tasks, the higher the similarity between two nodes in an ideal subspace, the greater the likelihood that they belong to the same cluster*, we expect the approximations of $z_x{z_x}^T$ and $z_a{z_a}^T$ to learn a better subspace for better similarity approximation with this design, guiding a better clustering results.
> > >
> > >
> > >
> > > In the revised manuscript, we have clarified the motivation for autoencoder like structure in Section 3.2 shown as $\color{darkred}{DarkRed}$.
> > >
> > > [r1] P. Zhu, X. Yao, Y. Wang, B. Hui, D. Du and Q. Hu. Multiview Deep Subspace Clustering Networks. in *IEEE Transactions on Cybernetics*, vol. 54, no. 7, pp. 4280-4293, July 2024.
> > >
> > > [r2] Xu J, Ren Y, Li G, et al. Deep embedded multi-view clustering with collaborative training[J]. Information Sciences, 2021, 573: 279-290.
> > >
> > > [r3] Peng X, Feng J, Xiao S, et al. Structured autoencoders for subspace clustering[J]. IEEE Transactions on Image Processing, 2018, 27(10): 5076-5086.
> > >
> > > [r4] Baldi P. Autoencoders, unsupervised learning, and deep architectures[C]//Proceedings of ICML workshop on unsupervised and transfer learning. JMLR Workshop and Conference Proceedings, 2012: 37-49.

---

> ### Author Response · Authors · 2024-12-06
> **Part2 Response**
>
> **#4** The definition of homophily (Definition 1) is similar to the concept of edge homophily from [r1], and this connection should be explicitly acknowledged or cited as the definition of homophily and heterophily is not the contribution of the current work. Furthermore, a more complexed scenario of homophily and heterophily is proposed in [r2], add discussion to clarify the scope of the current research.
>
> **[response]**: Thank you for your suggestion. As stated in the original manuscript, Definition 1 is the same as the homophily defined in [r1]. We restate it here to differentiate it from the definitions of homogeneity/heterogeneity and to avoid reader confusion. We have added the citation directly within the definition to make it easier for readers to trace the source. Compared to [r3], both definitions of homophily measure the graph's homophily by calculating the proportion of edges between nodes of the same class. The difference lies in that [r3] extends this definition to accommodate multi-label node classification tasks, whereas our work focuses on the multi-view heterophilous graph clustering problem and does not yet consider multi-label issues.
>
>
>
> **#5** The concept defined in Proposition 1 overlaps with "cross-class neighborhood similarity (CCNS)" introduced in [r3]. Discuss the differences or provide appropriate citations to avoid confusion.
>
> **[response]**: Thank you for your suggestion. Proposition 1 is derived from *CCNS*, and we have included a clear citation in the original manuscript. It is restated here to help readers understand our motivation and to support Hypothesis 1. To avoid any confusion, we have revised the wording as follows:
>
> > Following these previous works[r3], we emphasize their observation in the following Proposition 1.
>
> In addition, we highlight that Proposition 1 cannot be directly generalized to the focus of this paper, i.e., unsupervised MVHGC clustering task, so we need to generalize this proposition and utilize the neighbor pattern similarity by proposing Hypothesis 1.
>
>
>
> *[r1] Beyond Homophily in Graph Neural Networks: Current Limitations and Effective Designs*
> *Zhu et al., 2020.*
>
> *[r2] Multi-label Node Classification On Graph-Structured Data*
> *Zhao et al., 2023.*
>
> *[r3] Is Homophily a Necessity for Graph Neural Networks?*
> *Ma et al., 2021.*
>
>
>
> **#6** The loss terms $L{\text{sim}a}$ and $L{\text{sim}_x}$ aim to align $Z_aZ_a^T$ with $AA^T$ and $Z_xZ_x^T$ with $XX^T$. Would this lead the model to just use the original adjacency matrix ( A ) and feature matrix ( X )?
>
> **[response]**: The primary objective of $L{{sim}_a}$ and $L{{sim}_x}$ is not to replicate $A$ and $X$, but to guide the encoders $f_a$ and $f_x$ to capture the underlying good heterophily hidden within $A$ and $X$. The encoded representations $Z_a$ and $Z_x$ are optimized to better reflect these properties rather than simply mimicking the inputs. In addition, the effectiveness of this approach is empirically validated in our experiments, as shown in Table 2 and Figure 3. These results demonstrate that the similarity-regularized embeddings significantly enhance clustering performance compared to directly using $A$ and $X$. This confirms that the model learns improved representations rather than merely relying on the original inputs. We have included a further discussion about why we introduce such approximation of $Z_a$ and $Z_x$:
>
> > *The autoencoder design for projecting $\mathbf{A}$ to $\mathbf{Z_a}$ by $f_a(\cdot)$ and $\mathbf{X}$ to $\mathbf{Z_x}$ by $f_x(\cdot)$ has two main goals: (1) Multi-view collaborative optimization: the backpropagation of final multi-view clustering loss ($\mathcal{L}_{KL}$) introduces clustering information from other views [r1, r2], achieving multi-view collaborative optimization for each single view's similarity matrices. (2) Learn a better subspace: as stated in Hypothesis 1, the similarity matrices are expected to be constructed in an ideal subspace. The projection of $\mathbf{Z}_x \mathbf{Z}_x^T$ and $\mathbf{Z}_a \mathbf{Z}_a^T$ can not only preserve most important similarity information (implemented by regularization term $\mathcal{L}{sim}$) and feature information (by $\mathcal{L}_r$), but also obtain the guidance from other views [r3, r4], resulting in the possibility of learning a ideal subspace.*

---

> ### Author Response · Authors · 2024-12-06
> **Part3 Response**
>
> **#8** Describe the metrics used in this paper NMI and HR, including whether higher or lower values are better.
>
> **[response]**: Thank you for your advise. We've updated the manuscript to clarify the metrics. The modification is as follows:
>
> > NMI here is a metric used to assess the similarity between the true class labels and the predicted labels, where higher values indicate better clustering performance. Homophilous Ratio (named HR), refers to the proportion of edges within the same class compared to the total number of edges in the graph, which is used to measure the homophily of the graph.
>
>
>
> **#9** Explain the "parameter-free message passing layer" referenced in Figure 3.
>
> **[response]**: The parameter-free message passing layer refers to the message passing mechanism in Eq.(1), except that it does not include any learnable parameters, as stated in Par1 Response, **#3**.
>
> > **Model selection**: to exclude the influential factors from model design, optimization influence, we design a very basic GNN that contains only two non-parametric message passing layers. The basic GNN can be formulated as below. After $\mathbf{h}^2$ obtained, we conduct K-means to get final clustering results and compute its NMI scores.
>
>
>
>
> **#10** Elaborate on why the increase in homophilous information fails to improve clustering performance via a message-passing layer (e.g., the blue line in Figure 3).
>
> **[response]**: Thanks for your careful review. A direct response for this question is that "a message passing layer cannot capture the increased homophilous information of neighbor patterns". Specifically, we increase the 'homophilous  information' by increasing the HR of neighbor patterns, while the HR of original graph is kept low. Therefore, the clustering performance on original graphs does not improve due to the unchanged HR of original graph, although the homophilous information is added by increasing HR of neighbor patterns.
>
> This limitation highlights the need for a more sophisticated approach, such as our proposed similarity-enhanced framework, which directly extracts and utilizes homophilous information through similarity matrices before applying message passing. We will clarify this reasoning in the revised manuscript. Thank you for helping us improve our explanation. We have added this explaination to the revisions.
>
>
>
>
>
> **#11** Provide a detailed explanation of how ( P ) and ( Q ) are computed in Section 3.5.
>
> **[response]**: Thank you for your suggestion. The definitions of $P$ and $Q$ follows [r4]. In addition, we have added a more detailed explanation and relevant references in the Appendix D.1 to help readers better understand the purpose of this loss function.
>
> >In this multi-view clustering task, we apply Kullback-Leibler divergence loss ($\mathcal{L}kl$) to facilitate the model to obtain distinguishable embedding following the previous related works. Specifically, let $q_{ij}^v \in \mathbf{Q}^v$ be the soft cluster assignment that describes the possibility of node $i$ belongs to cluster centroid $j$ in $v$-th view and $\mathbf{P}$ be the sharpen target distribution, the final loss can be expressed as:}
> >$$
> >    \mathcal{L}{kl} = \text{KL}(\overline{\mathbf{P}} \Vert \overline{\mathbf{Q}}) + \sum_v^V \text{KL}(\overline{\mathbf{P}} \Vert \mathbf{Q}^v),
> >$$
> >
> >where $\overline{\mathbf{Q}}$ and $\overline{\mathbf{P}}$ represent the soft and target distribution of $\overline{\mathbf{H}}$, respectively. Following [r4], the soft assignment $q_{ij}$ and sharpen assignment $p_{ij}$ can be expressed by the utilization of Student's $t$-distribution as a kernel:
> >$$
> >q_{ij} = \frac{(1+ \Vert h_i - \mu_j \Vert ^2 / \alpha)^{-\frac{\alpha+1}{2}}}{\sum_{j'} 1+ \Vert h_i - \mu_{j'} \Vert ^2 / \alpha)^{-\frac{\alpha+1}{2}}}, \\
> >p_{ij} = \frac{q_{ij}^2 / \sum_iq_{ij}}{\sum_{j'}{(q_{ij'}^2 / \sum_iq_{ij'})}},
> >$$
> >where $h_i$ denotes the final embedding, $\mu_j$ is the centroid embedding of $j$-th cluster, and $\alpha$ denotes the degrees of freedom of the Student's $t$-distribution which is setting to $1$ for all experiments. With Eq.(3), we expect the first term to enhance the discriminability of the final embedding $\overline{\mathbf{H}}$, followed by the second term to encourage the soft distribution of each view to fit the distribution of $\overline{\mathbf{H}}$, and thus to exploit the inter-view consistency.}
>
>
> *[r4] J. Xie, R. Girshick, and A. Farhadi, “Unsupervised deep embedding for clustering analysis"*

---

### Author Response · Authors · 2024-12-06
**Thanks all reviwers for your constructive comments!**

We sincerely appreciate all reviewers for their insightful and constructive comments on our manuscript, which has greatly improved the quality of this work.

In the following, we respond to the reviewers' comments question-by-question, and the corresponding revisions in the manuscript are colored in  $\color{blue}{blue}$ for reviewer $\color{blue}{JU9m}$, $\color{darkgreen}{darkgreen}$ for reviewer $\color{darkgreen}{suHe}$, and $\color{darkred}{darkred}$ for reviewer $\color{darkred}{eMMK}$. $\color{red}{light red}$ colored revisions are for all reviewers.

---

### Decision · Action_Editor_rD99 · 2025-02-06

**Recommendation:** Reject

**Comment:**

The paper tackles the problem of unsupervised node classification with additional multi-view information. The authors proposed SMHGC, which integrates feature similarity, neighbor pattern similarity, and multi-view global similarity to generate node embeddings.

The major concerns after rebuttal from reviewers lies in the point that there are several design choice of the algorihtm not well justified. For example, the motivation for approximating the original similarity matrices remains unresolved, and the inter-view fusion approach is still unconvincing (Reviewer eMMK and JU9m).

**Audience:**

Yes. The paper is interested to graph learning community.

**Claims And Evidence:**

The paper conducted empirical comparison to justify the claim. However, the rationale of the components in algorithm design is not clear.

For example, as Reviewer eMMK pointed out, the rationale for final embedding H to compute weights or scores is only heuristic discussed, without empirical ablation study.

**Resubmission Of Major Revision:**

The authors may consider submitting a major revision at a later time.